# Replica method for eigenvalues of real Wishart product matrices

Jacob A. Zavatone-Veth[1,2]⋆ and Cengiz Pehlevan[3,2]⋆

**1** Department of Physics, Harvard University, Cambridge, MA, USA
**2** Center for Brain Science, Harvard University, Cambridge, MA, USA
**3** John A. Paulson School of Engineering and Applied Sciences, Harvard University, Cambridge, MA, USA
⋆ jzavatoneveth@g.harvard.edu, cpehlevan@seas.harvard.edu

January 24, 2023

## Abstract

**We show how the replica method can be used to compute the asymptotic eigenvalue spectrum of a real Wishart product matrix. For unstructured factors, this provides a compact, elementary derivation of a polynomial condition on the Stieltjes transform first proved by Müller [IEEE Trans. Inf. Theory. 48, 2086-2091 (2002)]. We then show how this computation can be extended to ensembles where the factors are drawn from matrix Gaussian distributions with general correlation structure. For both unstructured and structured ensembles, we derive polynomial conditions on the average values of the minimum and maximum eigenvalues, which in the unstructured case match the results obtained by Akemann, Ipsen, and Kieburg [Phys. Rev. E 88, 052118 (2013)] for the complex Wishart product ensemble.**

## 1   Introduction

In this note, we describe how the replica method from the statistical mechanics of disordered systems may be used to obtain the asymptotic density of eigenvalues for a Wishart product matrix

$$\mathbf{K} = \frac{1}{n_L \cdots n_1} \mathbf{X}_1^\top \cdots \mathbf{X}_L^\top \mathbf{X}_L \cdots \mathbf{X}_1, \tag{1}$$

where the factors

$$\mathbf{X}_\ell \in \mathbb{R}^{n_\ell \times n_{\ell-1}} \tag{2}$$

are independent Gaussian random matrices. In the simplest case, the factors are real Ginibre random matrices, i.e., they have independent and identically distributed standard real Gaussian elements $(X_\ell)_{ij} \sim \mathcal{N}(0,1)$, though the complex Gaussian case is also often studied [1–15]. We will also consider cases in which the elements of each factor are correlated.

Not all of our final results are novel. Rather, our overarching objective in reporting these replica-theoretic derivations are to note their simplicity, as the replica method has to the best of our knowledge not seen broad application to the study of product random matrices [9], despite its common usage in other areas of random matrix theory [16–22]. For a discussion of the application of the cavity method to Wishart product matrices, we direct the reader to the work of Dupic and Pérez Castillo [9], or to recent work by Cui, Rocks, and Mehta [23].

### 1.1   Applications of Wishart product matrices in science and technology

The spectral statistics of Wishart product matrices are of interest in many areas of physics and applied mathematics [7, 8]. For example, they describe the covariance statistics of Gaussian data

propagated through noisy linear vector channels [1]—in other words, the covariance statistics of certain linear latent variable models [24]—and transport in simple models for chaotic systems [11, 25]. Both real and complex Wishart product matrices are of particular interest in mathematical physics because certain features are amenable to exact study [2–13, 15].

Most commonly, Wishart product matrices are studied either at finite size or in one of three asymptotic limits. Adopting the nomenclature that the factor dimensions $n_\ell$ are the "widths" and the number of factors $L$ is the "depth" of the product, these limiting regimes are as follows:

- The thermodynamic limit, in which the widths are taken to infinity proportionally, i.e.,

$$n_0, \cdots, n_L \to \infty \quad \text{with} \quad \frac{n_\ell}{n_0} \to \alpha_\ell \in (0, \infty), \tag{3}$$

  for fixed depth $L$ [1–4, 8, 26, 27]. This is the regime on which we focus.

- The ergodic limit, in which the depth $L \to \infty$ for fixed widths $n_\ell$ [7, 8, 12, 26, 28].

- The double-scaling, or critical, regime, in which the depth $L$ and widths $n_\ell$ tend jointly to infinity [7, 8, 11, 15, 27–29].

Properties of the thermodynamic limit of real Wishart product matrices have recently attracted attention in the machine learning community, as they appear as the Neural Network Gaussian Process Kernel Gram matrix of a deep linear neural network with Gaussian inputs [23, 27, 30–34, 34–36]. In this case, $n_0$ represents the number of datapoints on which the kernel is evaluated, $n_1$ is the input dimensionality, and $n_2, \ldots, n_L$ are the widths of the hidden layers. The spectrum of this kernel matrix determines the generalization properties of a network in the limit of infinite hidden layer width [31–33]. The present note is based on our recent work on deep linear networks in Ref. [31]; we direct the interested reader to that work and references therein for more background on generalization in deep linear neural networks.

## 1.2 Roadmap

Our paper is organized as follows:

- In §2.1, we briefly introduce the Edwards-Jones [18] approach to computing the resolvent of a random matrix using the replica method.

- In §2.2, we apply the Edward-Jones method to compute the limiting spectral statistics of Wishart product matrices with uncorrelated factors. The details of this computation are deferred to Appendix A. This recovers a polynomial condition on the resolvent first proved by Müller [1].

- In §2.3, we extend this approach to structured Wishart product matrices where the factors have correlated rows and columns, deferring the details of the computation to Appendix B. We obtain a condition on the resolvent in terms of the spectral generating functions of the factor correlations, which to our knowledge as not previously been reported for $L > 1$ [37].

- In §3.1, we introduce the spherical spin glass method for computing the averages of the minimum and maximum eigenvalues of a random matrix using the replica trick [38–41].

- In §3.2, we apply this method to Wishart product matrices with uncorrelated factors, with the details of the computation given in Appendix C. The resulting polynomial conditions on the minimum and maximum eigenvalues match the results obtained by Akemann, Ipsen, and Kieburg [5] for the complex Wishart product ensemble.

- In §3.3, we extend this approach to ensembles with row-structured factors, deferring the details of the calculation to Appendix D. As in our analysis of the resolvent for structured ensembles, this result has to our knowledge not been previously reported for $L > 1$.

- In §4, we conclude by discussing the outlook for the application of the replica method to product matrix ensembles.

## 2  Replica approach to computing the resolvent

Before summarizing our results, let us briefly record our notational conventions. We denote vectors and matrices by bold lowercase and uppercase Roman letters, respectively, e.g., $\mathbf{x}$ and $\mathbf{X}$. For an integer $m$, $\mathbf{I}_m$ denotes the $m \times m$ identity matrix, while $\mathbf{1}_m$ denotes the $m$-dimensional vector with all elements equal to 1. We use $\propto$ to denote equality up to irrelevant constants of proportionality. Finally, we warn the reader that we will often leave implicit the domains of integrals.

### 2.1  The Edwards-Jones method for computing the resolvent

In the thermodynamic limit $n_0, \cdots, n_L \to \infty$, $n_\ell / n_0 \to \alpha_\ell \in (0, \infty)$, the eigenvalue density $\rho(\lambda)$ of $\mathbf{K}$ is self-averaging, and can be conveniently described in terms of its Stieltjes transform

$$G(z) = \lim_{n_0,\ldots,n_L \to \infty} \frac{1}{n_0} \text{tr}[(\mathbf{K} - z\mathbf{I}_{n_0})^{-1}], \tag{4}$$

from which the limiting density can be recovered via

$$\rho(\lambda) = \lim_{\epsilon \downarrow 0} \frac{1}{\pi} \text{Im}\, G(\lambda - i\epsilon). \tag{5}$$

To compute the the Stieltjes transform using the replica method from the statistical physics of disordered systems [16, 42, 43], we follow a standard approach, introduced by Edwards and Jones [18]. This method proceeds by writing

$$G(z) = \frac{\partial g}{\partial z} \tag{6}$$

for

$$g(z) = \lim_{n_0,\ldots,n_L \to \infty} \frac{2}{n_0} \log Z(z), \tag{7}$$

where the *partition function* is

$$Z(z) = \int_{\mathbb{R}^{n_0}} d\mathbf{w} \exp\left(-\frac{i}{2}\mathbf{w}^\top(z\mathbf{I}_{n_0} - \mathbf{K})\mathbf{w}\right). \tag{8}$$

In the thermodynamic limit, we expect $g(z)$ to be self-averaging, i.e., to concentrate around its expectation $\mathbb{E}g$ over the random factors $\mathbf{X}_\ell$. The expectation $\mathbb{E}\log Z$ can be evaluated using the identity $\mathbb{E}\log Z = \lim_{m\to 0} m^{-1}\log\mathbb{E}Z^m$ and a standard non-rigorous interchange of limits:

$$g = \lim_{n_0,\dots,n_L\to\infty} \frac{2}{n_0}\mathbb{E}\log Z = \lim_{m\to 0}\lim_{n_0,\dots,n_L\to\infty}\frac{2}{mn_0}\log\mathbb{E}Z^m. \tag{9}$$

As usual, we evaluate the moments $\mathbb{E}Z^m$ for non-negative integer $m$, and assume that they can be safely analytically continued to $m \to 0$ [42,43]. Here, as in other applications of the replica trick to the Stieltjes transform, the annealed average is exact, in the sense that the replica-symmetric saddle point is replica-diagonal [16–18] (see Appendices A and B).

## 2.2 Spectral moments for unstructured factors

In Ref. [1], Müller proved that the Stieltjes transform of a Wishart product matrix with unstructured factors (i.e., $(X_\ell)_{ij} \sim_{\text{i.i.d.}} \mathcal{N}(0,1)$) satisfies the polynomial equation

$$\frac{zG(z)+1}{G(z)} = \prod_{\ell=1}^{L}\left(1 - \frac{zG(z)+1}{\alpha_\ell}\right); \tag{10}$$

see also Refs. [3–5,9,10]. As noted by Burda *et al.* [3], the condition (10) can be expressed more compactly as

$$z = \frac{M(z)+1}{M(z)}\prod_{\ell=1}^{L}\left(1 + \frac{M(z)}{\alpha_\ell}\right) \tag{11}$$

in terms of the moment generating function

$$M(z) = \sum_{k=1}^{\infty}\frac{1}{z^k}\frac{1}{n_0}\mathrm{tr}(\mathbf{K}^k) = \frac{1}{n_0}\mathrm{tr}[(z\mathbf{I}_{n_0} - \mathbf{K})^{-1}\mathbf{K}] = -zG(z) - 1, \tag{12}$$

where we assume that the formal series converges. Our first result is a derivation, presented in Appendix A, of (10) using the Edwards-Jones method outlined in §2.1.

In the case $L = 1$, the equation for the Stieltjes transform reduces to

$$\frac{zG(z)+1}{G(z)} = 1 - \frac{zG(z)+1}{\alpha_1} \tag{13}$$

which can be re-written as

$$0 = z + \frac{1}{G(z)} - \frac{\alpha_1}{\alpha_1 + G(z)} \tag{14}$$

which is the familiar result for a Wishart matrix. In the equal-width case $\alpha_1 = \cdots = \alpha_L = \alpha$, we have the simplification

$$\frac{zG(z)+1}{G(z)} = \left(1 - \frac{zG(z)+1}{\alpha}\right)^L. \tag{15}$$

In the context of deep linear neural networks, this special case has a natural interpretation as a network with hidden layer widths equal to the input dimensions. If $L = 2$, this is a cubic equation,

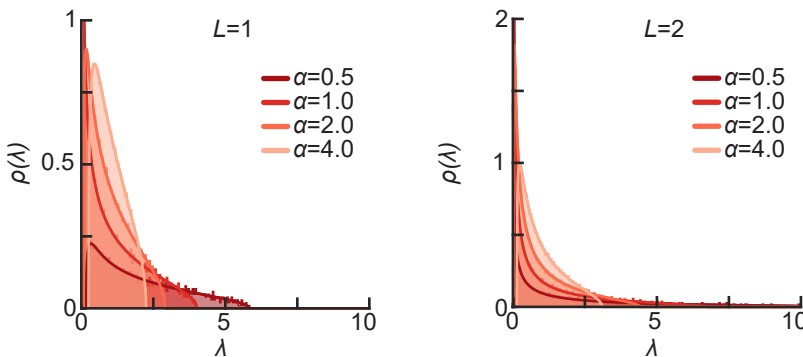

Figure 1: Eigenvalue densities for unstructured Wishart product matrices for depths $L = 1$ (*left*) and $L = 2$ (*right*) for varying widths $\alpha_1 = \cdots = \alpha_L = \alpha$, indicated by shades of red. Solid lines show the result of solving equation (15) numerically, while shaded areas show the results of numerical eigendecompositions of matrices of size $n_0 = 2048$. Importantly, each empirical histogram is obtained for a single realization of the random matrix.

which can be solved in radicals, though the result is not particularly illuminating [9, 23]. In the square case $\alpha_1 = \cdots = \alpha_L = 1$, we have the further simplification

$$0 = z^L G(z)^{L+1} - z G(z) - 1. \tag{16}$$

As shown in previous works, this can be solved to obtain an exact expression for the eigenvalue density [10]. More generally, the equation (10) must be solved numerically. We show examples for $L = 1$ and $L = 2$ in Figure 1, demonstrating excellent agreement with numerical experiment. We direct the reader to previous work by Burda *et al.* [3] and by Dupic and Pérez Castillo [9] for further examples.

## 2.3 Spectral moments for structured factors

Importantly, the replica approach is not limited to the study of ensembles where the factors have independent and identically distributed entries. It also allows one to tackle with relative ease the more general setting where the factors are independent matrix Gaussian random variables, i.e.,

$$\mathbb{E}[(X_\ell)_{ij}] = 0, \tag{17}$$
$$\mathbb{E}[(X_\ell)_{ij}(X_\ell)_{kl}] = (\Sigma_\ell)_{ik}(\Gamma_\ell)_{jl} \tag{18}$$

for row-wise covariance matrices

$$\Sigma_\ell \in \mathbb{R}^{n_\ell \times n_\ell} \tag{19}$$

and column-wise covariance matrices

$$\Gamma_\ell \in \mathbb{R}^{n_{\ell-1} \times n_{\ell-1}}. \tag{20}$$

For the thermodynamic limit to be well-defined, we have in mind an ensemble defined by sequences of covariance matrices $\Sigma_\ell(n_\ell)$, $\Gamma_\ell(n_{\ell-1})$ such that the bulk spectral statistics of these matrices tend to deterministic limits (see Appendix B for a more precise statement of our assumptions on these matrices).

We can equivalently define this ensemble by

$$\mathbf{K} = \frac{1}{n_L \cdots n_1} \mathbf{\Gamma}_1^{1/2} \mathbf{Z}_1^\top \mathbf{\Sigma}_1^{1/2} \cdots \mathbf{\Gamma}_L^{1/2} \mathbf{Z}_L^\top \mathbf{\Sigma}_L \mathbf{Z}_L \mathbf{\Gamma}_L^{1/2} \cdots \mathbf{\Sigma}_1^{1/2} \mathbf{Z}_1 \mathbf{\Gamma}_1^{1/2} \tag{21}$$

for $\mathbf{Z}_\ell$ an unstructured Ginibre matrix with standard Gaussian elements $(Z_\ell)_{ij} \sim \mathcal{N}(0,1)$. This re-writing makes it clear that we may take the columns of all factors except $\mathbf{X}_1$ to be uncorrelated without loss of generality, as the ensemble with

$$\mathbb{E}[(X_\ell)_{ij}] = 0, \tag{22}$$
$$\mathbb{E}[(X_\ell)_{ij}(X_\ell)_{kl}] = (\tilde{\Sigma}_\ell)_{ik}\delta_{jl} \qquad (\ell = 2, \ldots, L) \tag{23}$$
$$\mathbb{E}[(X_1)_{ij}(X_1)_{kl}] = (\tilde{\Sigma}_1)_{ik}(\Gamma_1)_{jl} \tag{24}$$

for

$$\tilde{\mathbf{\Sigma}}_\ell = \mathbf{\Sigma}_\ell^{1/2} \mathbf{\Gamma}_{\ell+1} \mathbf{\Sigma}_\ell^{1/2} \tag{25}$$

is identically distributed, where we write $\mathbf{\Gamma}_{L+1} = \mathbf{I}_{n_L}$ for brevity. As a result, we henceforth set

$$\mathbf{\Gamma}_\ell = \mathbf{I}_{n_{\ell-1}} \quad \text{for} \quad \ell = 2, \ldots, L, \tag{26}$$

hence $\tilde{\mathbf{\Sigma}}_\ell = \mathbf{\Sigma}_\ell$ for all $\ell = 1, \ldots, L$. Moreover, we may take the covariance matrices $\mathbf{\Sigma}_\ell$ to be diagonal without loss of generality, as the random Gaussian factors are rotation-invariant. In the case where the columns of the first factor are uncorrelated, i.e., $\mathbf{\Gamma}_1 = \mathbf{I}_{n_0}$, then the ensemble is rotation-invariant, and one can consider only row structure without loss of generality. This ensemble describes the kernel of a deep linear neural network with independent input examples, or more generally the covariance of a linear latent variable model [24].

For matrices from this correlated ensemble, we show in Appendix B that the moment generating function $M(z)$ of $\mathbf{K}$ satisfies the self-consistent equation

$$z = M_{\mathbf{\Gamma}_1}^{-1}(M(z)) \prod_{\ell=1}^{L} \left[ \frac{M(z)}{\alpha_\ell} M_{\mathbf{\Sigma}_\ell}^{-1} \left( \frac{M(z)}{\alpha_\ell} \right) \right]. \tag{27}$$

Here, the functions

$$M_{\mathbf{\Sigma}_\ell}(z) = \lim_{n_\ell \to \infty} \frac{1}{n_\ell} \text{tr}[(z\mathbf{I}_{n_\ell} - \mathbf{\Sigma}_\ell)^{-1} \mathbf{\Sigma}_\ell] \tag{28}$$

are the moment generating functions of the matrices $\mathbf{\Sigma}_\ell$, and the inverse functions $M_{\mathbf{\Sigma}_\ell}^{-1}(z)$ satisfy $(M_{\mathbf{\Sigma}_\ell}^{-1} \circ M_{\mathbf{\Sigma}_\ell})(z) = z$. We can re-write this as

$$M(z) = M_{\mathbf{\Gamma}_1} \left( \frac{z}{\prod_{\ell=1}^{L} \left[ \frac{M(z)}{\alpha_\ell} M_{\mathbf{\Sigma}_\ell}^{-1} \left( \frac{M(z)}{\alpha_\ell} \right) \right]} \right). \tag{29}$$

This condition can of course be equivalently written in terms of the resolvent $G(z)$. Moreover, the inverses of the spectral generating functions can be equivalently expressed in terms of the $S$-transform from free probability theory [26].

In the case $L = 1$, this ensemble reduces to the ordinary correlated Wishart ensemble [37, 44, 45], and (27) recapitulates the result previously obtained by Burda *et al.* [37]. However, the general $L > 1$ case does not appear to have been reported in the literature [24, 37, 44–46].

In the case in which the first factor has uncorrelated columns, i.e., $\mathbf{\Gamma}_1 = \mathbf{I}_{n_0}$, we have

$$M_{\mathbf{\Gamma}_1}(z) = \frac{1}{z - 1} \tag{30}$$

and

$$M_{\mathbf{\Gamma}_1}^{-1}(z) = 1 + \frac{1}{z}, \tag{31}$$

hence we obtain the simplified condition

$$z = \frac{M(z) + 1}{M(z)} \prod_{\ell=1}^{L} \left[ \frac{M(z)}{\alpha_\ell} M_{\mathbf{\Sigma}_\ell}^{-1} \left( \frac{M(z)}{\alpha_\ell} \right) \right]. \tag{32}$$

If $L = 1$, we can further simplify this condition to

$$M_{\mathbf{\Sigma}_1} \left( \frac{\alpha_1 z}{M(z) + 1} \right) = \frac{M(z)}{\alpha_1}, \tag{33}$$

recapitulating the result of Burda *et al.* [37]. It is easy to confirm that this result reduces to that which we obtained before for the unstructured case. For $\mathbf{\Sigma}_\ell = \mathbf{I}_{n_\ell}$, we have

$$M_{\mathbf{\Sigma}_\ell}(z) = \frac{1}{z - 1} \tag{34}$$

and

$$M_{\mathbf{\Sigma}_\ell}^{-1}(z) = 1 + \frac{1}{z}, \tag{35}$$

hence (32) reduces to (11). Another simplifying case is when all layers are identically structured, i.e., $M_{\mathbf{\Sigma}_1}(z) = \cdots = M_{\mathbf{\Sigma}_L}(z)$, and the widths are equal, i.e., $\alpha_1 = \cdots = \alpha_L = \alpha$. Then, we have the simplified condition

$$M_{\mathbf{\Sigma}_1} \left( \frac{\alpha}{M(z)} \left( \frac{M(z)z}{1 + M(z)} \right)^{1/L} \right) = \frac{M(z)}{\alpha}. \tag{36}$$

To gain intuition for how the structured case differs from the unstructured setting, we consider a simple example. With the application of neural network kernels in mind, we include structured correlations only in $\mathbf{X}_1$, corresponding to the case in which the dataset is composed of independent samples drawn from a Gaussian distribution with correlated dimensions. We keep the remaining factors unstructured—i.e., $\mathbf{\Sigma}_\ell = \mathbf{I}_{n_\ell}$ for $\ell = 2, \ldots, L$—corresponding to a setting in which the weights of the network are drawn independently. This is the standard setting for deep linear neural networks, where the weights at initialization are assumed to be independent and identically distributed [27, 30, 32, 33].

As a toy model for structured data, we consider a gapped model in which a fraction $\gamma \in [0, 1]$ of the eigenvalues of $\mathbf{\Sigma}_1$ are equal to $\sigma > 1$, while the remainder are equal to unity. In the case

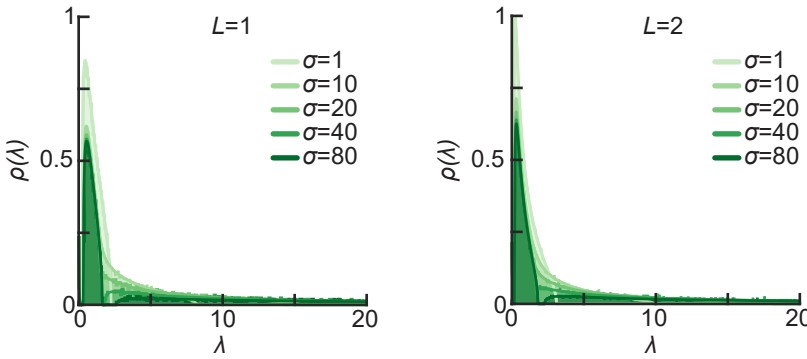

Figure 2: Eigenvalue densities for structured Wishart product matrices for depths $L = 1$ (*left*) and $L = 2$ (*right*) of width $\alpha_1 = \cdots = \alpha_L = \alpha = 4$. The correlation structure is as described in the main text, with $M_{\Sigma_1}(z)$ given by (37) with $\gamma = 1/8$ and varying signal eigenvalues $\sigma$, indicated by shades of green. Solid lines show the result of solving equation (38) numerically, while shaded areas show the results of numerical eigendecompositions of matrices of size $n_0 = 2048$. Importantly, each empirical histogram is obtained for a single realization of the random matrix.

$\gamma = 0$, this reduces to the unstructured spectrum considered before. For simplicity, we restrict our attention to equal-width factors $\alpha_1 = \cdots = \alpha_L = \alpha$. With this setup, we have

$$M_{\Sigma_1}(z) = \gamma \frac{\sigma}{z - \sigma} + (1 - \gamma)\frac{1}{z - 1}, \tag{37}$$

and the simplified condition on the generating function

$$M_{\Sigma_1}\left(\frac{\alpha z}{(1 + M(z))(1 + M(z)/\alpha)^{L-1}}\right) = \frac{M(z)}{\alpha}. \tag{38}$$

We show examples of this model for $L = 1$ and $L = 2$ in Figure 2, demonstrating excellent agreement with numerical experiment. As the signal eigenvalue $\sigma$ increases, we see that the bulk density separates into two components. It will be interesting to investigate this effect, and other effects of structured correlations, in future work.

For this simple data model, we can also study the case in which all layers include identical structure, i.e., $M_{\Sigma_1}(z) = M_2(z) = \cdots = M_{\Sigma_L}(z)$. In the equal-width case $\alpha_1 = \cdots = \alpha_L = \alpha$, this gives the simplified condition noted above in (36). In Figure 3, we compare the results of solving (36) for this model to numerical experiments, showing excellent agreement. Interestingly, in this case the gap in the spectrum that is present for $L = 1$ (for which this model is identical to that considered above and in Figure 2) is not present at $L = 2$.

# 3 Replica approach to computing the extremal eigenvalues

## 3.1 The spherical spin glass method for computing extremal eigenvalues

In the thermodynamic limit, we expect the typical minimum and maximum eigenvalues of $\mathbf{K}$, which define the edges of the bulk spectrum, to be self-averaging. Conditions on these eigenvalues

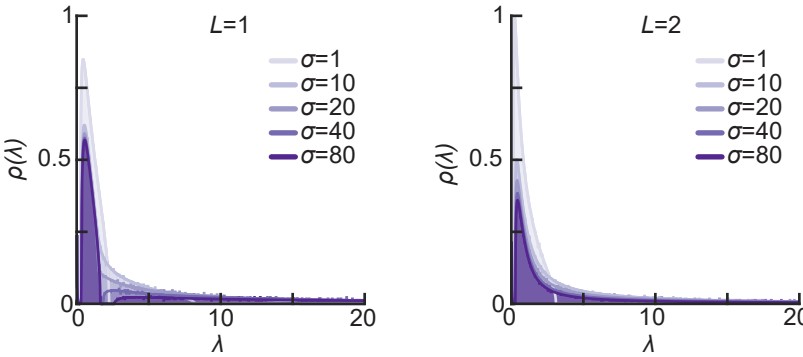

Figure 3: Eigenvalue densities for structured Wishart product matrices for depths $L = 1$ (*left*) and $L = 2$ (*right*) of width $\alpha_1 = \cdots = \alpha_L = \alpha = 4$. The correlation structure is as described in the main text, with $M_{\Sigma_1}(z) = \cdots = M_{\Sigma_L}(z)$ given by (37) with $\gamma = 1/8$ and varying signal eigenvalues $\sigma$, indicated by shades of purple. Solid lines show the result of solving equation (36) numerically, while shaded areas show the results of numerical eigendecompositions of matrices of size $n_0 = 2048$. Importantly, each empirical histogram is obtained for a single realization of the random matrix.

can be obtained from the condition (10) on the Stieltjes transform (see Ref. [5]), but they can also be computed using a direct, physically meaningful method.

In this approach, the eigenvalues are interpreted as the ground-state energies of a spherical spin glass, as studied by Kosterlitz, Thouless, and Jones [38], and in subsequent random matrix theory works [39–41]. Our starting point is the min-max characterization of the minimum and maximum eigenvalues as Rayleigh quotients:

$$\lambda_{\min}(\mathbf{K}) = \min_{\mathbf{w} \in \mathbb{R}^{n_0}, \|\mathbf{w}\|=1} \mathbf{w}^\top \mathbf{K} \mathbf{w}, \qquad \lambda_{\max}(\mathbf{K}) = \max_{\mathbf{w} \in \mathbb{R}^{n_0}, \|\mathbf{w}\|=1} \mathbf{w}^\top \mathbf{K} \mathbf{w}. \tag{39}$$

We first consider the computation of the minimimum eigenvalue. We introduce a Gibbs distribution at inverse temperature $\beta > 0$ over vectors in the sphere $\mathbb{S}^{n_0-1}(\sqrt{n_0})$ of radius $\sqrt{n_0}$ in $n_0$ dimensions, with density

$$p(\mathbf{w}; \beta, \mathbf{K}) = \frac{1}{Z(\beta, \mathbf{K})} \exp[-\beta E(\mathbf{w}, \mathbf{K})] \tag{40}$$

with respect to the Lebesgue measure on the sphere. Here,

$$E(\mathbf{w}, \mathbf{K}) = \frac{1}{2} \mathbf{w}^\top \mathbf{K} \mathbf{w} \tag{41}$$

is the energy function associated to the minimization problem (39), and the partition function is

$$Z(\beta, \mathbf{K}) = \int_{\mathbb{S}^{n_0-1}(\sqrt{n_0})} d\mathbf{w} \, \exp[-\beta E(\mathbf{w}, \mathbf{K})]. \tag{42}$$

As $\beta \to \infty$, the Gibbs distribution (40) will concentrate on the ground state of (41), which is the eigenvector of $\mathbf{K}$ corresponding to its minimum eigenvalue. We denote averages with respect to

the Gibbs distribution (40) by $\langle \cdot \rangle_{\beta,\mathbf{K}}$. Then, recalling our definition of $E$ in (41) and the Rayleigh quotient (39), we have

$$\mathbb{E}\lambda_{\min}(\mathbf{K}) = \lim_{\beta \to \infty} \mathbb{E}\frac{2}{n_0}\langle E \rangle_{\beta,\mathbf{K}} = \lim_{\beta \to \infty} \frac{\partial g(\beta, \mathbf{K})}{\partial \beta}, \tag{43}$$

where we have defined the *reduced free energy per site*

$$g(\beta, \mathbf{K}) = -\frac{2}{n_0}\mathbb{E}\log Z(\beta, \mathbf{K}). \tag{44}$$

In the thermodynamic limit, we expect $\log Z$ to be self-averaging, and it can be computed using the replica method.

We can also use this setup to compute the minimum eigenvalue. We can see that this computation is identical up to a sign, and that

$$\mathbb{E}\lambda_{\max}(\mathbf{K}) = -\lim_{\beta \to \infty} \mathbb{E}\frac{2}{n_0}\langle E \rangle_{-\beta,\mathbf{K}} = \lim_{\beta \to \infty} \frac{\partial g(-\beta, \mathbf{K})}{\partial \beta}. \tag{45}$$

As the rank of $\mathbf{K}$ is at most $\min\{n_0, \ldots, n_L\}$,

$$\lambda_{\min}(\mathbf{K}) = 0 \quad \text{if} \quad \min\{\alpha_1, \ldots, \alpha_L\} < 1. \tag{46}$$

If $\min\{\alpha_1, \ldots, \alpha_L\} > 1$, then we expect the minimum eigenvalue to be almost surely positive.

## 3.2  Extremal eigenvalues for unstructured factors

As in our study of the Stieltjes transform, we first consider an ensemble with unstructured factors, i.e., $(X_\ell)_{ij} \sim_{\text{i.i.d.}} \mathcal{N}(0,1)$. Deferring the details of the replica computation to §C, we find that the edges of the spectrum can be written as

$$\mathbb{E}\lambda_{\min/\max} = \left(1 + \frac{1}{A}\right)\prod_{\ell=1}^{L}\left(1 + \frac{A}{\alpha_\ell}\right) \tag{47}$$

where $A$ is a solution to the equation

$$A = \frac{1}{\sum_{\ell=1}^{L}\frac{A}{\alpha_\ell + A}} - 1. \tag{48}$$

This computation is somewhat more tedious than that of the Stieltjes transform, as the replica-symmetric saddle point is not replica-diagonal.

These conditions are identical to those obtained by Akemann, Ipsen, and Kieburg [5] for the complex Wishart ensemble. In general, one must determine which of the solutions to these equations give the edges of the spectrum. However, as noted by Akemann, Ipsen, and Kieburg [5], they are exactly solvable in the equal-width case $\alpha_1 = \cdots = \alpha_L = \alpha$. With this constraint, $A$ is determined by the quadratic equation $LA^2 + (L-1)A - \alpha = 0$, which gives

$$A = \frac{\sqrt{4L\alpha + (L-1)^2} - (L-1)}{2L}, \tag{49}$$

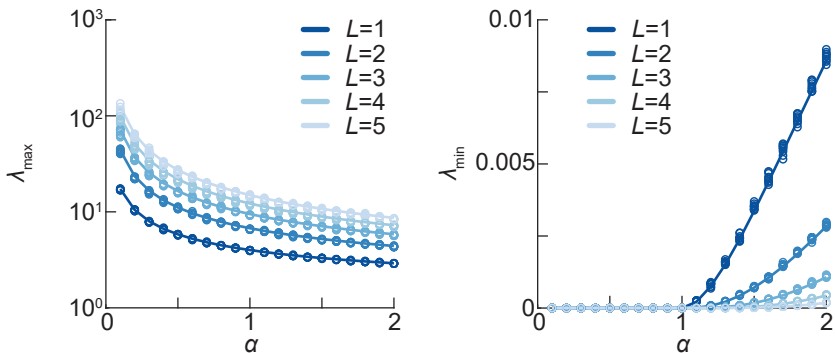

Figure 4: Maximum (*left*) and minimum (*right*) eigenvalues of Wishart product matrices for varying depths $L$ (with higher values indicated by lighter shades of blue) and varying widths $\alpha_1 = \alpha_2 = \cdots = \alpha_L = \alpha$. In each panel, the theoretical predictions from equations (50) and (52), respectively, are plotted as solid lines, while the open circles show the results of numerical eigendecompositions for 10 realizations of matrices of size $n_0 = 1000$.

and thus

$$\mathbb{E}\lambda_{\max} = \frac{\alpha + A}{\alpha - (L-1)A}\left(1 + \frac{A}{\alpha}\right)^L. \tag{50}$$

Considering the minimum eigenvalue, we have the quadratic equation $LB^2 - (L-1)B - \alpha = 0$ for $B = -A$, which yields

$$B = \frac{\sqrt{4L\alpha + (L-1)^2} + (L-1)}{2L}, \tag{51}$$

and thus

$$\mathbb{E}\lambda_{\min} = \frac{\alpha - B}{\alpha + (L-1)B}\left(1 - \frac{B}{\alpha}\right)^L. \tag{52}$$

If $L = 1$, this recovers the familiar results for Wishart matrices. In the square case $\alpha = 1$, we have the further simplification

$$\mathbb{E}\lambda_{\max} = (L+1)\left(1 + \frac{1}{L}\right)^L, \tag{53}$$

while $\lambda_{\min} = 0$, as noted previously by Dupic and Pérez Castillo [9]. In Figure 4, we show that these results display excellent agreement with numerical eigendecompositions.

## 3.3 Extremal eigenvalues for factors with correlated rows

As in our analysis of the resolvent in §2.3, we can extend the computation of the extremal eigenvalues to ensembles with correlated factors. For the sake of simplicity, we focus on ensembles with only row-wise structure, i.e.,

$$\mathbb{E}[(X_\ell)_{ij}] = 0, \tag{54}$$
$$\mathbb{E}[(X_\ell)_{ij}(X_\ell)_{kl}] = (\Sigma_\ell)_{ik}\delta_{jl}. \tag{55}$$

As discussed in §2.3, this restriction can be made without loss of generality so long as $\Gamma_1 = \mathbf{I}_{n_0}$. We provide further discussion of why this restriction simplifies the computation in §D; briefly, it is compatible with the spherical constraint.

Then, deferring the details of the computation to §D, we find that the edges of the spectrum are determined by

$$\mathbb{E}\lambda_{\min/\max} = \left(1 + \frac{1}{A}\right)\prod_{\ell=1}^{L} \frac{A}{\alpha_\ell} M_{\Sigma_\ell}^{-1}\left(\frac{A}{\alpha_\ell}\right), \tag{56}$$

where $A$ is a solution of

$$A = \frac{1}{\sum_{\ell=1}^{L}(\mu_\ell(A)-1)/\mu_\ell(A)} - 1, \tag{57}$$

for

$$\mu_\ell(A) = -\frac{\alpha_\ell}{A}\frac{M_{\Sigma_\ell}^{-1}(A/\alpha_\ell)}{(M_{\Sigma_\ell}^{-1})'(A/\alpha_\ell)}. \tag{58}$$

Here, $(M_{\Sigma_\ell}^{-1})'$ denotes the first derivative of $M_{\Sigma_\ell}^{-1}$ with respect to its argument; $\mu_\ell$ is therefore proportional to the multiplicative inverse of the logarithmic derivative of $M_{\Sigma_\ell}^{-1}$. As in the unstructured case, one must determine which of the solutions to (57) give the edges of the spectrum [5].

In the unstructured case, we have

$$\mu_\ell(z) = 1 + \frac{z}{\alpha_\ell} \tag{59}$$

for $\ell = 1, \dots, L$, hence we recover the result of §3.2.

To demonstrate the effects of structure, we revisit the equal-width model with structure in the first layer and no structure elsewhere, as introduced in §2.3. In this case, we have $\mu_\ell(z) = 1 + z/\alpha$ for $\ell = 2, \dots, L$, hence we have the simplified equation

$$\mathbb{E}\lambda_{\min/\max} = \left(1 + \frac{1}{A}\right)\frac{A}{\alpha}M_{\Sigma_1}^{-1}\left(\frac{A}{\alpha}\right)\left(1 + \frac{A}{\alpha}\right)^{L-1}, \tag{60}$$

where $A$ is a solution of

$$[(LA + \alpha)\mu_1(A) - (\alpha + A)](1 + A) = (\alpha + A)\mu_1(A). \tag{61}$$

In Figure 5, we show that this result agrees with with numerical eigendecompositions for matrices with varying fraction of signal eigenvalues $\gamma$.

# 4   Conclusion

We have shown that the replica method affords a useful approach to the study of product random matrices. These derivations are straightforward, but they are of course not mathematically rigorous [42, 43]. We conclude by briefly discussing the utility of these results vis-à-vis open questions in the study of product random matrices.

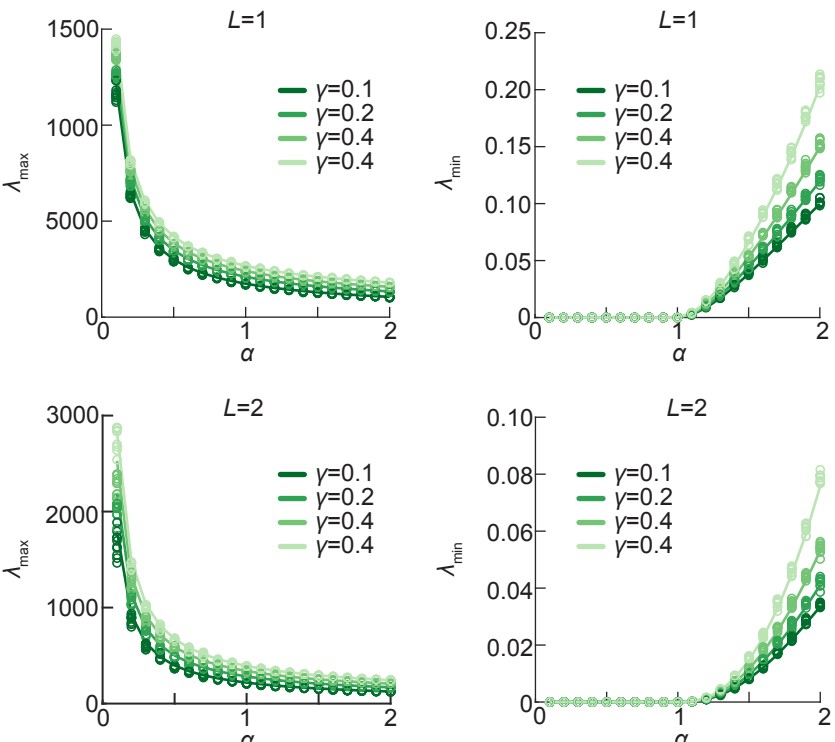

Figure 5: Maximum (*left*) and minimum (*right*) eigenvalues for structured Wishart product matrices for depths $L = 1$ (*top*) and $L = 2$ (*bottom*) of varying widths $\alpha_1 = \cdots = \alpha_L = \alpha$. The correlation structure is as described in the main text, with $M_{\Sigma_1}(z)$ given by (37) with signal eigenvalues $\sigma = 100$ and varying signal fraction $\gamma$, indicated by shades of green. Solid lines show the result of solving equation (60) numerically, while the open circles show the results of numerical eigendecompositions for 10 realizations of matrices of size $n_0 = 1000$.

The most notable utility of statistical physics methods, including the replica trick, in random matrix theory is that they allow for the study of non-invariant ensembles. Dating back to the seminal work of Bray and Rogers [17, 47], sparse ensembles have been of particular interest [9, 17, 41]. We hope that the methods described in this work will enable further investigation of products of sparse random matrices and of other non-invariant product ensembles. It will also be interesting to investigate Gaussian ensembles with general correlations between the factor matrices [9, 37, 45, 46]. We remark that the approaches used in this work are particularly simple due to the independence of different factors, i.e., $\mathbb{E}[(X_\ell)_{ij}(X_{\ell'})_{kl}] = 0$ if $\ell \neq \ell'$, hence studying ensembles with correlated factors would require a somewhat different replica-theoretic setup.

In the context of neural networks, the structured ensemble with row-wise correlations studied in this work has a natural interpretation as the neural network Gaussian process kernel of a deep linear network where the features and input dimensions are correlated but the datapoints are independent samples. It will be interesting to study the spectra of such kernel matrices in greater detail in future work. To enable future studies of generalization in deep nonlinear random feature models and wide neural networks [31, 48], it will be important to extend to extend these approaches to the nonlinear setting [34–36, 49]. Finally, it will be interesting to investigate the

spectra resulting from the non-Gaussian factor distributions that arise in trained Bayesian neural networks [32, 33].

# Acknowledgements

We are indebted to Gernot Akemann for his helpful comments, and for drawing our attention to recent work on the double-scaling regime. We thank Boris Hanin for inspiring discussions. We also thank Blake Bordelon for comments on an early version of this manuscript. Finally, we thank the referees for their useful suggestions.

**Author contributions**    JAZ-V conceived the project, performed all research, and wrote the paper. CP supervised the project and contributed to review and editing.

**Funding information**    JAZ-V and CP were supported by a Google Faculty Research Award and NSF DMS-2134157.

# A    Computing the Stieltjes transform for unstructured factors

In this appendix, we derive the result (10) for the Stieltjes transform of a Wishart product matrix with unstructured factors. Our starting point is the partition function (8) in the Edwards-Jones [18] approach. We divide the details of the derivation into two parts. In §A.1, we evaluate the moments of the partition function. Then, in §A.2, we derive the replica-symmetric saddle point equations and use them to obtain the desired condition on $G(z)$ and $M(z)$.

## A.1    Step I: Evaluating the moments of the partition function

Introducing replicas indexed by $a = 1, \ldots, m$, the moments of the partition function (8) expand as

$$\mathbb{E} Z^m = \int \prod_{a=1}^m d\mathbf{w}^a \, \exp\left(-\frac{iz}{2}\sum_{a=1}^m \|\mathbf{w}^a\|^2\right) \mathbb{E} \exp\left(\frac{i}{2n_L \cdots n_1}\sum_{a=1}^m (\mathbf{w}^a)^\top \mathbf{X}_1^\top \cdots \mathbf{X}_L^\top \mathbf{X}_L \cdots \mathbf{X}_1 \mathbf{w}^a\right). \quad (62)$$

Using the fact that the rows of $\mathbf{X}_L$ are independent and identically distributed standard Gaussian random vectors in $\mathbb{R}^{n_{L-1}}$, we have

$$\mathbb{E}_{\mathbf{X}_L} \exp\left(\frac{i}{2n_L \cdots n_1}\sum_{a=1}^m (\mathbf{w}^a)^\top \mathbf{X}_1^\top \cdots \mathbf{X}_L^\top \mathbf{X}_L \cdots \mathbf{X}_1 \mathbf{w}^a\right) \quad (63)$$

$$= \det\left(\mathbf{I}_{n_{L-1}} - \frac{i}{n_L \cdots n_1}\sum_{a=1}^m \mathbf{X}_{L-1} \cdots \mathbf{X}_1 \mathbf{w}^a (\mathbf{w}^a)^\top \mathbf{X}_1^\top \cdots \mathbf{X}_{L-1}^\top\right)^{-n_L/2} \quad (64)$$

$$= \det(\mathbf{I}_m - \mathbf{C}_L)^{-n_L/2} \quad (65)$$

where in the last line we have applied the Weinstein–Aronszajn identity to express the determinant in terms of the Wick-rotated overlap matrix

$$C_L^{ab} \equiv \frac{i}{n_L \cdots n_1}(\mathbf{w}^a)^\top \mathbf{X}_1^\top \cdots \mathbf{X}_{L-1}^\top \mathbf{X}_{L-1} \cdots \mathbf{X}_1 \mathbf{w}^b. \quad (66)$$

We enforce the definition of these order parameters using Fourier representations of the $\delta$-distribution with corresponding Lagrange multipliers $\hat{C}_L^{ab}$, writing

$$
1 = \int \frac{d\mathbf{C}_L\, d\hat{\mathbf{C}}_L}{(4\pi i/n_L)^{m(m+1)/2}} \exp\left(-\frac{n_L}{2}\operatorname{tr}(\mathbf{C}_L\hat{\mathbf{C}}_L)\right)
$$
$$
\times \exp\left(\frac{i}{2n_{L-1}\cdots n_1}\sum_{a,b=1}^m \hat{C}_L^{ab}(\mathbf{w}^a)^\top\mathbf{X}_1^\top\cdots\mathbf{X}_{L-1}^\top\mathbf{X}_{L-1}\cdots\mathbf{X}_1\mathbf{w}^b\right). \tag{67}
$$

Here, the integrals over $\mathbf{C}_L$ are taken over $m\times m$ imaginary symmetric matrices, while the integrals over $\hat{\mathbf{C}}_L$ are taken over imaginary symmetric matrices. This yields

$$
\mathbb{E}Z^m = \int \frac{d\mathbf{C}_L\, d\hat{\mathbf{C}}_L}{(4\pi i/n_L)^{m(m+1)/2}} \exp\left(-\frac{n_L}{2}[\operatorname{tr}(\mathbf{C}_L\hat{\mathbf{C}}_L)+\log\det(\mathbf{I}_m-\mathbf{C}_L)]\right)
$$
$$
\times \int \prod_{a=1}^m d\mathbf{w}^a \exp\left(-\frac{iz}{2}\sum_{a=1}^m \|\mathbf{w}^a\|^2\right)
$$
$$
\times \mathbb{E}_{\mathbf{X}_1,\dots,\mathbf{X}_{L-1}} \exp\left(\frac{i}{2n_{L-1}\cdots n_1}\sum_{a,b=1}^m \hat{C}_L^{ab}(\mathbf{w}^a)^\top\mathbf{X}_1^\top\cdots\mathbf{X}_{L-1}^\top\mathbf{X}_{L-1}\cdots\mathbf{X}_1\mathbf{w}^b\right). \tag{68}
$$

We can easily see that $\mathbf{X}_{L-1}$ may be integrated out using a similar procedure, and that this may be iterated backwards by introducing order parameters

$$
C_\ell^{ab} \equiv \frac{i}{n_1\cdots n_\ell}(\mathbf{w}^a)^\top\mathbf{X}_1^\top\cdots\mathbf{X}_{\ell-1}^\top\mathbf{X}_{\ell-1}\cdots\mathbf{X}_1\mathbf{w}^b, \tag{69}
$$

yielding

$$
\mathbb{E}Z^m = \int \frac{d\mathbf{C}_1\, d\hat{\mathbf{C}}_1}{(4\pi i/n_1)^{m(m+1)/2}}\cdots\int \frac{d\mathbf{C}_L\, d\hat{\mathbf{C}}_L}{(4\pi i/n_L)^{m(m+1)/2}}
$$
$$
\exp\left(-\frac{1}{2}\sum_{\ell=1}^L n_\ell[\operatorname{tr}(\mathbf{C}_\ell\hat{\mathbf{C}}_\ell)+\log\det(\mathbf{I}_m-\mathbf{C}_\ell\hat{\mathbf{C}}_{\ell+1})]\right)
$$
$$
\times \int \prod_{a=1}^m d\mathbf{w}^a \exp\left(-\frac{iz}{2}\sum_{a=1}^m \|\mathbf{w}^a\|^2 + \frac{i}{2}\sum_{a,b=1}^m \hat{C}_1^{ab}(\mathbf{w}^a)^\top\mathbf{w}^b\right), \tag{70}
$$

where we have defined $\hat{\mathbf{C}}_{L+1}\equiv\mathbf{I}_m$ for brevity. We then can evaluate the remaining Gaussian integral over $\mathbf{w}^a$:

$$
\int \prod_{a=1}^m d\mathbf{w}^a \exp\left(-\frac{iz}{2}\sum_{a=1}^m \|\mathbf{w}^a\|^2 + \frac{i}{2}\sum_{a,b=1}^m \hat{C}_1^{ab}(\mathbf{w}^a)^\top\mathbf{w}^b\right) \tag{71}
$$
$$
= \int \prod_{a=1}^m d\mathbf{w}^a \exp\left(-\frac{i}{2}\sum_{a,b=1}^m (z\delta_{ab}-\hat{C}_1^{ab})(\mathbf{w}^a)^\top\mathbf{w}^b\right) \tag{72}
$$
$$
\propto \det(\hat{\mathbf{C}}_1-z\mathbf{I}_m)^{-n_0/2}. \tag{73}
$$

where we discard an irrelevant constant of proportionality. Therefore, we have

$$
\mathbb{E}Z^m \propto \int \frac{d\mathbf{C}_1\, d\hat{\mathbf{C}}_1}{(4\pi i/n_1)^{m(m+1)/2}}\cdots\int \frac{d\mathbf{C}_L\, d\hat{\mathbf{C}}_L}{(4\pi i/n_L)^{m(m+1)/2}} \exp\left(-\frac{n_0 m}{2}S(\mathbf{C}_1,\hat{\mathbf{C}}_1,\dots,\mathbf{C}_L,\hat{\mathbf{C}}_L)\right) \tag{74}
$$

for

$$S(\mathbf{C}_1, \hat{\mathbf{C}}_1, \ldots, \mathbf{C}_L, \hat{\mathbf{C}}_L) = \frac{1}{m} \log \det(\hat{\mathbf{C}}_1 - z\mathbf{I}_m)$$

$$+ \frac{1}{m} \sum_{\ell=1}^{L} \alpha_\ell [\text{tr}(\mathbf{C}_\ell \hat{\mathbf{C}}_\ell) + \log \det(\mathbf{I}_m - \mathbf{C}_\ell \hat{\mathbf{C}}_{\ell+1})] \tag{75}$$

where we recall the definition $\hat{\mathbf{C}}_{L+1} \equiv \mathbf{I}_m$. In the thermodynamic limit $n_0, n_1, \ldots, n_L \to \infty$, this integral can be evaluated using the method of steepest descent, yielding

$$-\frac{2}{n_0} \mathbb{E} \log Z = \underset{\mathbf{C}_1, \hat{\mathbf{C}}_1, \ldots, \mathbf{C}_L, \hat{\mathbf{C}}_L}{\text{extr}} S, \tag{76}$$

where the notation extr means that $S$ should be evaluated at the saddle point

$$\frac{\partial S}{\partial \mathbf{C}_\ell} = \frac{\partial S}{\partial \hat{\mathbf{C}}_\ell} = \mathbf{0} \qquad (\ell = 1, \ldots, L). \tag{77}$$

## A.2 Step II: The replica-symmetric saddle point equations

As is standard in the replica method (see e.g. Ref. [42]), we will consider replica-symmetric (RS) saddle points, where the order parameters take the form

$$\mathbf{C}_\ell = q_\ell \mathbf{I}_m + c_\ell \mathbf{1}_m \mathbf{1}_m^\top, \tag{78}$$

$$\hat{\mathbf{C}}_\ell = \hat{q}_\ell \mathbf{I}_m + \hat{c}_\ell \mathbf{1}_m \mathbf{1}_m^\top. \tag{79}$$

Under this *Ansatz*, we will now simplify $S$ in the limit $m \to 0$ using standard identities (see Ref. [31] and Refs. [42, 43]). We have

$$\lim_{m \to 0} \frac{1}{m} \text{tr}(\mathbf{C}_\ell \hat{\mathbf{C}}_\ell) = \lim_{m \to 0} [q_\ell \hat{q}_\ell + q_\ell \hat{c}_\ell + c_\ell \hat{q}_\ell + m c_\ell \hat{c}_\ell] \tag{80}$$

$$= q_\ell \hat{q}_\ell + q_\ell \hat{c}_\ell + c_\ell \hat{q}_\ell. \tag{81}$$

Using the matrix determinant lemma, we have

$$\lim_{m \to 0} \frac{1}{m} \log \det(\hat{\mathbf{C}}_1 - z\mathbf{I}_m) = \lim_{m \to 0} \frac{1}{m} \log \det[(\hat{q}_1 - z)\mathbf{I}_m + \hat{c}_1 \mathbf{1}_m \mathbf{1}_m^\top] \tag{82}$$

$$= \log(\hat{q}_1 - z) + \lim_{m \to 0} \frac{1}{m} \log\left(1 + \frac{m\hat{c}_1}{\hat{q}_1 - z}\right) \tag{83}$$

$$= \log(\hat{q}_1 - z) + \frac{\hat{c}_1}{\hat{q}_1 - z}, \tag{84}$$

and, similarly,

$$\lim_{m \to 0} \frac{1}{m} \log \det(\mathbf{I}_m - \mathbf{C}_\ell \hat{\mathbf{C}}_{\ell+1}) = \log(1 - q_\ell \hat{q}_{\ell+1}) - \frac{q_\ell \hat{c}_{\ell+1} + c_\ell \hat{q}_{\ell+1}}{1 - q_\ell \hat{q}_{\ell+1}}. \tag{85}$$

This gives

$$\lim_{m \to 0} S = \log(\hat{q}_1 - z) + \frac{\hat{c}_1}{\hat{q}_1 - z}$$

$$+ \sum_{\ell=1}^{L} \alpha_\ell \left( q_\ell \hat{q}_\ell + q_\ell \hat{c}_\ell + c_\ell \hat{q}_\ell + \log(1 - q_\ell \hat{q}_{\ell+1}) - \frac{q_\ell \hat{c}_{\ell+1} + c_\ell \hat{q}_{\ell+1}}{1 - q_\ell \hat{q}_{\ell+1}} \right) \tag{86}$$

with the boundary condition $\hat{q}_{L+1} = 1$, $\hat{c}_{L+1} = 0$.

We also have

$$G(z) = -\lim_{m \to 0} \frac{\partial S}{\partial z} = -\frac{1}{z - \hat{q}_1} - \frac{\hat{c}_1}{(z - \hat{q}_1)^2}, \tag{87}$$

where the order parameters are to be evaluated at their saddle point values.

From the equation $\partial S / \partial q_\ell = 0$, we have

$$0 = \hat{q}_\ell + \hat{c}_\ell - \frac{\hat{q}_{\ell+1}}{1 - q_\ell \hat{q}_{\ell+1}} - \frac{\hat{c}_{\ell+1}}{1 - q_\ell \hat{q}_{\ell+1}} - \frac{q_\ell \hat{c}_{\ell+1} + c_\ell \hat{q}_{\ell+1}}{(1 - q_\ell \hat{q}_{\ell+1})^2} \hat{q}_{\ell+1} \tag{88}$$

for $\ell = 1, \ldots, L$. From the equations $\partial S / \partial \hat{q}_\ell = 0$, we have

$$0 = -\frac{1}{z - \hat{q}_1} - \frac{\hat{c}_1}{(z - \hat{q}_1)^2} + \alpha_1 (q_1 + c_1) \tag{89}$$

if $\ell = 1$, and

$$0 = \alpha_\ell (q_\ell + c_\ell) - \alpha_{\ell-1} \left( \frac{q_{\ell-1}}{1 - q_{\ell-1}\hat{q}_\ell} + \frac{c_{\ell-1}}{1 - q_{\ell-1}\hat{q}_\ell} + \frac{q_{\ell-1}\hat{c}_\ell + c_{\ell-1}\hat{q}_\ell}{(1 - q_{\ell-1}\hat{q}_\ell)^2} q_{\ell-1} \right) \tag{90}$$

if $\ell > 1$. From $\partial S / \partial c_\ell = 0$, we have

$$0 = \hat{q}_\ell - \frac{\hat{q}_{\ell+1}}{1 - q_\ell \hat{q}_{\ell+1}} \tag{91}$$

for $\ell = 1, \ldots, L$. Finally, from $\partial S / \partial \hat{c}_\ell = 0$ we have

$$0 = -\frac{1}{z - \hat{q}_1} + \alpha_1 q_1 \tag{92}$$

if $\ell = 1$ and

$$0 = \alpha_\ell q_\ell - \alpha_{\ell-1} \frac{q_{\ell-1}}{1 - q_{\ell-1}\hat{q}_\ell} \tag{93}$$

for $\ell > 1$.

Simplifying, we find that the replica-nonuniform components are determined by the system

$$\hat{q}_\ell = \frac{\hat{q}_{\ell+1}}{1 - q_\ell \hat{q}_{\ell+1}} \qquad (\ell = 1, \ldots, L) \tag{94}$$

$$q_1 = \frac{1}{\alpha_1} \frac{1}{z - \hat{q}_1} \tag{95}$$

$$q_\ell = \frac{\alpha_{\ell-1}}{\alpha_\ell} \frac{q_{\ell-1}}{1 - q_{\ell-1}\hat{q}_\ell} \qquad (\ell = 2, \ldots, L), \tag{96}$$

while the uniform components are determined by

$$\hat{c}_\ell = \frac{\hat{c}_{\ell+1} + c_\ell \hat{q}_{\ell+1}^2}{(1 - q_\ell \hat{q}_{\ell+1})^2} \qquad (\ell = 1, \ldots, L) \tag{97}$$

$$c_1 = \frac{1}{\alpha_1} \frac{\hat{c}_1}{(z - \hat{q}_1)^2} \tag{98}$$

$$c_\ell = \frac{\alpha_{\ell-1}}{\alpha_\ell} \frac{c_{\ell-1} + q_{\ell-1}^2 \hat{c}_\ell}{(1 - q_{\ell-1}\hat{q}_\ell)^2} \qquad (\ell = 2, \ldots, L) \tag{99}$$

Recalling the boundary condition $\hat{q}_{L+1} = 1$, $\hat{c}_{L+1} = 0$, it is easy to see that we should have $c_\ell = \hat{c}_\ell = 0$ for all $\ell = 1, \ldots, L$. Then, the Stieltjes transform is given by $G(z) = -\alpha_1 q_1$, where $q_1$ is determined by the system

$$\hat{q}_\ell = \frac{\hat{q}_{\ell+1}}{1 - q_\ell \hat{q}_{\ell+1}} \qquad (\ell = 1, \ldots, L) \tag{100}$$

$$q_1 = \frac{1}{\alpha_1} \frac{1}{z - \hat{q}_1} \tag{101}$$

$$q_\ell = \frac{\alpha_{\ell-1}}{\alpha_\ell} \frac{q_{\ell-1}}{1 - q_{\ell-1} \hat{q}_\ell} \qquad (\ell = 2, \ldots, L). \tag{102}$$

These equations can be simplified with a bit of algebra, as in our prior work [31]. From the equation

$$\hat{q}_\ell = \frac{\hat{q}_{\ell+1}}{1 - q_\ell \hat{q}_{\ell+1}}, \tag{103}$$

we have

$$q_\ell = \frac{\hat{q}_\ell - \hat{q}_{\ell+1}}{\hat{q}_\ell \hat{q}_{\ell+1}} \tag{104}$$

for $\ell = 1, \ldots, L$. Then, for $\ell = 2, \ldots, L$, the equation

$$q_\ell = \frac{\alpha_{\ell-1}}{\alpha_\ell} \frac{q_{\ell-1}}{1 - q_{\ell-1} \hat{q}_\ell} \tag{105}$$

yields

$$\frac{\hat{q}_\ell - \hat{q}_{\ell+1}}{\hat{q}_\ell \hat{q}_{\ell+1}} = \frac{\alpha_{\ell-1}}{\alpha_\ell} \frac{\hat{q}_{\ell-1}}{\hat{q}_\ell} \frac{\hat{q}_{\ell-1} - \hat{q}_\ell}{\hat{q}_{\ell-1} \hat{q}_\ell}. \tag{106}$$

If we define $A$ by

$$\alpha_1 q_1 \hat{q}_1 = A \tag{107}$$

such that

$$\frac{\hat{q}_1 - \hat{q}_2}{\hat{q}_1 \hat{q}_2} = \frac{A}{\alpha_1 \hat{q}_1}, \tag{108}$$

we have

$$\frac{\hat{q}_2 - \hat{q}_3}{\hat{q}_2 \hat{q}_3} = \frac{\alpha_1}{\alpha_2} \frac{\hat{q}_1}{\hat{q}_2} \frac{\hat{q}_1 - \hat{q}_2}{\hat{q}_1 \hat{q}_2} \tag{109}$$

$$= \frac{A}{\alpha_2 \hat{q}_2}. \tag{110}$$

It is then easy to see that

$$q_\ell = \frac{\hat{q}_\ell - \hat{q}_{\ell+1}}{\hat{q}_\ell \hat{q}_{\ell+1}} = \frac{A}{\alpha_\ell \hat{q}_\ell} \tag{111}$$

for $\ell = 1, \ldots, L$. This yields the backward recurrence

$$\hat{q}_\ell = \left(1 + \frac{A}{\alpha_\ell}\right)\hat{q}_{\ell+1} \tag{112}$$

for $\ell = 1, \ldots, L$, which can be solved using the endpoint condition $\hat{q}_{L+1} = 1$, yielding

$$\hat{q}_\ell = \prod_{j=\ell}^{L}\left(1 + \frac{A}{\alpha_j}\right). \tag{113}$$

Then, using the fact that $q_1$ and $\hat{q}_1$ are related by the equation

$$q_1 = \frac{1}{\alpha_1}\frac{1}{z - \hat{q}_1}, \tag{114}$$

we have

$$\hat{q}_1 = -\frac{1 - z\alpha_1 q_1}{\alpha_1 q_1}, \tag{115}$$

so

$$A = -(1 - z\alpha_1 q_1). \tag{116}$$

Therefore, we have the equation

$$-\frac{1 - z\alpha_1 q_1}{\alpha_1 q_1} = \prod_{\ell=1}^{L}\left(1 - \frac{1 - z\alpha_1 q_1}{\alpha_\ell}\right) \tag{117}$$

which, substituting in $G(z) = -\alpha_1 q_1$, yields the condition

$$\frac{zG(z) + 1}{G(z)} = \prod_{\ell=1}^{L}\left(1 - \frac{zG(z) + 1}{\alpha_\ell}\right) \tag{118}$$

on the Stieltjes transform. This is the result claimed in (10).

# B  Computing the Stieltjes transform for structured factors

In this appendix, we derive the result (27) for the Stieltjes transform of a Wishart product matrix with correlated factors. This computation parallels our analysis of the unstructured case in §A. Again, our starting point is the Edwards-Jones [18] partition function, and we once again first evaluate its moments in §B.1 and then derive and simplify the replica-symmetric saddle point equations in §B.2.

## B.1  Step I: Evaluating the moments of the partition function

Introducing replicas indexed by $a = 1, \ldots, m$, the moments of the partition function (8) expand as

$$\mathbb{E}Z^m = \int \prod_{a=1}^{m} d\mathbf{w}^a \exp\left(-\frac{iz}{2}\sum_{a=1}^{m}\|\mathbf{w}^a\|^2\right)\mathbb{E}\exp\left(\frac{i}{2n_L \cdots n_1}\sum_{a=1}^{m}(\mathbf{w}^a)^\top \mathbf{X}_1^\top \cdots \mathbf{X}_L^\top \mathbf{X}_L \cdots \mathbf{X}_1 \mathbf{w}^a\right). \tag{119}$$

We will first integrate out $\mathbf{X}_L$. For brevity, define the matrix $\mathbf{A}_L \in \mathbb{R}^{n_{L-1} \times m}$ by

$$(A_L)_{ja} = \frac{1}{\sqrt{n_L \cdots n_1}}(\mathbf{X}_{L-1} \cdots \mathbf{X}_1 \mathbf{w}^a)_j \tag{120}$$

such that the required expectation is

$$\mathbb{E}_{\mathbf{X}_L} \exp\left(\frac{i}{2n_L \cdots n_1} \sum_{a=1}^{m} (\mathbf{w}^a)^\top \mathbf{X}_1^\top \cdots \mathbf{X}_L^\top \mathbf{X}_L \cdots \mathbf{X}_1 \mathbf{w}^a\right) = \mathbb{E}_{\mathbf{X}_L} \exp\left(\frac{i}{2} \text{tr}[\mathbf{A}_L^\top \mathbf{X}_L^\top \mathbf{X}_L \mathbf{A}_L]\right). \tag{121}$$

Let $\mathbf{Z}_L \in \mathbb{R}^{n_L \times n_{L-1}}$ be a real Ginibre random matrix, with independent and identically distributed elements $Z_{ij} \sim \mathcal{N}(0,1)$, such that

$$\mathbf{X}_L = \boldsymbol{\Sigma}_L^{1/2} \mathbf{Z}_L \boldsymbol{\Gamma}_L^{1/2} \tag{122}$$

in distribution. Then, we can easily evaluate the expectation using column-major vectorization:

$$\mathbb{E}_{\mathbf{X}_L} \exp\left(\frac{i}{2} \text{tr}[\mathbf{A}_L^\top \mathbf{X}_L^\top \mathbf{X}_L \mathbf{A}_L]\right) = \mathbb{E}_{\mathbf{Z}_L} \exp\left(\frac{i}{2} \text{tr}[\mathbf{Z}_L^\top \boldsymbol{\Sigma}_L \mathbf{Z}_L \boldsymbol{\Gamma}_L^{1/2} \mathbf{A}_L \mathbf{A}_L^\top \boldsymbol{\Gamma}_L^{1/2}]\right) \tag{123}$$

$$= \mathbb{E}_{\mathbf{Z}_L} \exp\left(\frac{i}{2} \text{vec}(\mathbf{Z}_L)^\top [(\boldsymbol{\Gamma}_L^{1/2} \mathbf{A}_L \mathbf{A}_L^\top \boldsymbol{\Gamma}_L^{1/2}) \otimes \boldsymbol{\Sigma}_L] \text{vec}(\mathbf{Z}_L)\right) \tag{124}$$

$$= \det(\mathbf{I}_{n_L n_{L-1}} - i(\boldsymbol{\Gamma}_L^{1/2} \mathbf{A}_L \mathbf{A}_L^\top \boldsymbol{\Gamma}_L^{1/2}) \otimes \boldsymbol{\Sigma}_L)^{-1/2}, \tag{125}$$

where $\text{vec}(\cdot)$ denotes the column-major vectorization of a matrix and $\otimes$ denotes the Kronecker product [50]. Using the mixed-product property of the Kronecker product and the Weinstein–Aronszajn identity [50, 51], we have

$$\det(\mathbf{I}_{n_L n_{L-1}} - i(\boldsymbol{\Gamma}_L^{1/2} \mathbf{A}_L \mathbf{A}_L^\top \boldsymbol{\Gamma}_L^{1/2}) \otimes \boldsymbol{\Sigma}_L) = \det\left(\mathbf{I}_{n_L n_{L-1}} - i(\boldsymbol{\Gamma}_L^{1/2} \mathbf{A}_L \otimes \boldsymbol{\Sigma}_L)(\mathbf{A}_L^\top \boldsymbol{\Gamma}_L^{1/2} \otimes \mathbf{I}_{n_L})\right) \tag{126}$$

$$= \det\left(\mathbf{I}_{mn_L} - i(\mathbf{A}_L^\top \boldsymbol{\Gamma}_L^{1/2} \otimes \mathbf{I}_{n_L})(\boldsymbol{\Gamma}_L^{1/2} \mathbf{A}_L \otimes \boldsymbol{\Sigma}_L)\right) \tag{127}$$

$$= \det(\mathbf{I}_{mn_L} - i(\mathbf{A}_L^\top \boldsymbol{\Gamma}_L \mathbf{A}_L) \otimes \boldsymbol{\Sigma}_L). \tag{128}$$

We now introduce the Wick-rotated order parameters

$$C_L^{ab} \equiv i(\mathbf{A}_L^\top \boldsymbol{\Gamma}_L \mathbf{A}_L)_{ab} = \frac{i}{n_L \cdots n_1}(\mathbf{w}^a)^\top \mathbf{X}_1^\top \cdots \mathbf{X}_{L-1}^\top \boldsymbol{\Gamma}_L \mathbf{X}_{L-1} \cdots \mathbf{X}_1 \mathbf{w}^b, \tag{129}$$

which differs from the order parameters used in our previous computation due to the inclusion of the column correlation matrix $\boldsymbol{\Gamma}_\ell$.

We enforce the definition of these order parameters using Fourier representations of the $\delta$-distribution with corresponding Lagrange multipliers $\hat{C}_L^{ab}$, which gives

$$\mathbb{E}Z^m = \int \frac{d\mathbf{C}_L \, d\hat{\mathbf{C}}_L}{(4\pi i/n_L)^{m(m+1)/2}} \exp\left(-\frac{n_L}{2}\left[\text{tr}(\mathbf{C}_L \hat{\mathbf{C}}_L) + \frac{1}{n_L} \log \det(\mathbf{I}_{mn_L} - \mathbf{C}_L \otimes \boldsymbol{\Sigma}_\mathbf{L})\right]\right)$$

$$\times \int \prod_{a=1}^{m} d\mathbf{w}^a \exp\left(-\frac{iz}{2} \sum_{a=1}^{m} \|\mathbf{w}^a\|^2\right)$$

$$\times \mathbb{E}_{\mathbf{X}_1,\dots,\mathbf{X}_{L-1}} \exp\left(\frac{i}{2n_{L-1} \cdots n_1} \sum_{a,b=1}^{m} \hat{C}_L^{ab}(\mathbf{w}^a)^\top \mathbf{X}_1^\top \cdots \mathbf{X}_{L-1}^\top \boldsymbol{\Gamma}_L \mathbf{X}_{L-1} \cdots \mathbf{X}_1 \mathbf{w}^b\right). \tag{130}$$

We now integrate out $\mathbf{X}_{L-1}$. Define the $n_{L-2} \times m$ matrix

$$(A_{L-1})_{ja} = \frac{1}{\sqrt{n_{L-1} \cdots n_1}} (\mathbf{X}_{L-2} \cdots \mathbf{X}_1 \mathbf{w}^a)_j, \tag{131}$$

such that

$$\mathbb{E}_{\mathbf{X}_{L-1}} \exp\left( \frac{i}{2n_{L-1} \cdots n_1} \sum_{a,b=1}^{m} \hat{C}_L^{ab} (\mathbf{w}^a)^\top \mathbf{X}_1^\top \cdots \mathbf{X}_{L-1}^\top \mathbf{\Gamma}_L \mathbf{X}_{L-1} \cdots \mathbf{X}_1 \mathbf{w}^b \right) \tag{132}$$

$$= \mathbb{E}_{\mathbf{X}_{L-1}} \exp\left( \frac{i}{2} \operatorname{tr}[\hat{\mathbf{C}}_L \mathbf{A}_{L-1}^\top \mathbf{X}_{L-1}^\top \mathbf{\Gamma}_L \mathbf{X}_{L-1} \mathbf{A}_{L-1}] \right) \tag{133}$$

$$= \mathbb{E}_{\mathbf{Z}_{L-1}} \exp\left( \frac{i}{2} \operatorname{tr}[\hat{\mathbf{C}}_L \mathbf{A}_{L-1}^\top \mathbf{\Gamma}_{L-1}^{1/2} \mathbf{Z}_{L-1}^\top \mathbf{\Sigma}_{L-1}^{1/2} \mathbf{\Gamma}_L \mathbf{\Sigma}_{L-1}^{1/2} \mathbf{Z}_{L-1} \mathbf{\Gamma}_{L-1}^{1/2} \mathbf{A}_{L-1}] \right) \tag{134}$$

where we write

$$\mathbf{X}_{L-1} = \mathbf{\Sigma}_{L-1}^{1/2} \mathbf{Z}_{L-1} \mathbf{\Gamma}_{L-1}^{1/2} \tag{135}$$

for a standard $n_{L-1} \times n_{L-2}$ Ginibre random matrix $\mathbf{Z}_{L-1}$. Defining the matrix

$$\tilde{\mathbf{\Sigma}}_{L-1} \equiv \mathbf{\Sigma}_{L-1}^{1/2} \mathbf{\Gamma}_L \mathbf{\Sigma}_{L-1}^{1/2} \in \mathbb{R}^{n_{L-1} \times n_{L-1}}, \tag{136}$$

we can see that we can evaluate and simplify the expectation over $\mathbf{Z}_{L-1}$ in the same way as we did the expectation over $\mathbf{Z}_L$, yielding

$$\det(\mathbf{I}_{mn_{L-1}} - i(\hat{\mathbf{C}}_L \mathbf{A}_{L-1}^\top \mathbf{\Gamma}_{L-1} \mathbf{A}_{L-1}) \otimes \tilde{\mathbf{\Sigma}}_{L-1})^{-1/2}. \tag{137}$$

This can in turn be written in terms of the order parameters

$$\mathbf{C}_{L-1} = i \mathbf{A}_{L-1}^\top \mathbf{\Gamma}_{L-1} \mathbf{A}_{L-1}. \tag{138}$$

Then, as in the unstructured case, we can see that we can iterate this procedure backward by introducing order parameters

$$C_\ell^{ab} \equiv \frac{i}{n_1 \cdots n_\ell} (\mathbf{w}^a)^\top \mathbf{X}_1^\top \cdots \mathbf{X}_{\ell-1}^\top \mathbf{\Gamma}_\ell \mathbf{X}_{\ell-1} \cdots \mathbf{X}_1 \mathbf{w}^b, \tag{139}$$

yielding

$$\mathbb{E}Z^m = \int \frac{d\mathbf{C}_1 \, d\hat{\mathbf{C}}_1}{(4\pi i/n_1)^{m(m+1)/2}} \cdots \int \frac{d\mathbf{C}_L \, d\hat{\mathbf{C}}_L}{(4\pi i/n_L)^{m(m+1)/2}}$$

$$\exp\left( -\frac{1}{2} \sum_{\ell=1}^{L} n_\ell \left[ \operatorname{tr}(\mathbf{C}_\ell \hat{\mathbf{C}}_\ell) + \frac{1}{n_\ell} \log \det[\mathbf{I}_{mn_\ell} - (\mathbf{C}_\ell \hat{\mathbf{C}}_{\ell+1}) \otimes \tilde{\mathbf{\Sigma}}_\ell] \right] \right)$$

$$\times \int \prod_{a=1}^{m} d\mathbf{w}^a \exp\left( -\frac{iz}{2} \sum_{a=1}^{m} \|\mathbf{w}^a\|^2 + \frac{i}{2} \sum_{a,b=1}^{m} \hat{C}_1^{ab} (\mathbf{w}^a)^\top \mathbf{\Gamma}_1 \mathbf{w}^b \right), \tag{140}$$

where for the sake of brevity we have defined

$$\tilde{\mathbf{\Sigma}}_\ell = \mathbf{\Sigma}_\ell^{1/2} \mathbf{\Gamma}_{\ell+1} \mathbf{\Sigma}_\ell^{1/2} \tag{141}$$

for $\ell = 1, \ldots, L-1$ and

$$\tilde{\Sigma}_L = \Sigma_L, \tag{142}$$

and $\hat{\mathbf{C}}_{L+1} \equiv \mathbf{I}_m$. The integral over $\mathbf{w}^a$ is now once again a matrix Gaussian, and yields

$$\det(\hat{\mathbf{C}}_1 \otimes \Gamma_1 - z\mathbf{I}_{mn_0})^{-1/2} \tag{143}$$

up to an irrelevant constant of proportionality. Therefore, we obtain

$$\mathbb{E}Z^m \propto \int \frac{d\mathbf{C}_1 \, d\hat{\mathbf{C}}_1}{(4\pi i/n_1)^{m(m+1)/2}} \cdots \int \frac{d\mathbf{C}_L \, d\hat{\mathbf{C}}_L}{(4\pi i/n_L)^{m(m+1)/2}} \exp\left(-\frac{n_0 m}{2} S(\mathbf{C}_1, \hat{\mathbf{C}}_1, \ldots, \mathbf{C}_L, \hat{\mathbf{C}}_L)\right) \tag{144}$$

for

$$\begin{aligned}
S(\mathbf{C}_1, \hat{\mathbf{C}}_1, \ldots, \mathbf{C}_L, \hat{\mathbf{C}}_L) = {} & \frac{1}{mn_0} \det(\hat{\mathbf{C}}_1 \otimes \Gamma_1 - z\mathbf{I}_{mn_0}) \\
& + \frac{1}{m} \sum_{\ell=1}^{L} \alpha_\ell \left[ \mathrm{tr}(\mathbf{C}_\ell \hat{\mathbf{C}}_\ell) + \frac{1}{n_\ell} \log \det[\mathbf{I}_{mn_\ell} - (\mathbf{C}_\ell \hat{\mathbf{C}}_{\ell+1}) \otimes \tilde{\Sigma}_\ell] \right],
\end{aligned} \tag{145}$$

where we recall the definition $\hat{\mathbf{C}}_{L+1} \equiv \mathbf{I}_m$.

In the thermodynamic limit, we expect that

$$\frac{1}{n_\ell} \log \det[\mathbf{I}_{mn_\ell} - (\mathbf{C}_\ell \hat{\mathbf{C}}_{\ell+1}) \otimes \Sigma_\ell] \sim \mathcal{O}(1), \tag{146}$$

provided that the spectrum of $\Sigma_\ell$ is sufficiently generic. It clearly holds in the unstructured case $\Sigma_\ell = \sigma_\ell \mathbf{I}_{n_\ell}$, in which we have

$$\frac{1}{n_\ell} \log \det[\mathbf{I}_{mn_\ell} - (\mathbf{C}_\ell \hat{\mathbf{C}}_{\ell+1}) \otimes \Sigma_\ell] = \log \det[\mathbf{I}_m - \sigma_\ell \mathbf{C}_\ell \hat{\mathbf{C}}_{\ell+1}]. \tag{147}$$

Under the assumption that this scaling is valid, we can evaluate the required integrals using the method of steepest descent.

## B.2  Step II: The replica-symmetric saddle point equations

We now make an RS *Ansatz*

$$\mathbf{C}_\ell = q_\ell \mathbf{I}_m + c_\ell \mathbf{1}_m \mathbf{1}_m^\top, \tag{148}$$

$$\hat{\mathbf{C}}_\ell = \hat{q}_\ell \mathbf{I}_m + \hat{c}_\ell \mathbf{1}_m \mathbf{1}_m^\top. \tag{149}$$

The fist set of new terms relative to our calculation in the unstructured case are

$$\frac{1}{mn_\ell} \log \det[\mathbf{I}_{mn_\ell} - (\mathbf{C}_\ell \hat{\mathbf{C}}_{\ell+1}) \otimes \tilde{\Sigma}_\ell]. \tag{150}$$

More generally, we have

$$\det[\mathbf{I}_{mn_\ell} - (\mathbf{C}_\ell \hat{\mathbf{C}}_{\ell+1}) \otimes \tilde{\Sigma}_\ell] \tag{151}$$

$$= \det[\mathbf{I}_{mn_\ell} - q_\ell \hat{q}_{\ell+1} \mathbf{I}_m \otimes \tilde{\Sigma}_\ell - (q_\ell \hat{c}_{\ell+1} + c_\ell \hat{q}_{\ell+1} + mc_\ell \hat{c}_{\ell+1})(\mathbf{1}_m \mathbf{1}_m^\top) \otimes \tilde{\Sigma}_\ell] \tag{152}$$

$$= \det[\mathbf{I}_m \otimes (\mathbf{I}_{n_\ell} - q_\ell \hat{q}_{\ell+1} \tilde{\Sigma}_\ell) - (q_\ell \hat{c}_{\ell+1} + c_\ell \hat{q}_{\ell+1} + mc_\ell \hat{c}_{\ell+1})(\mathbf{1}_m \otimes \Sigma_\ell)(\mathbf{1}_m^\top \otimes \mathbf{I}_{n_\ell})] \tag{153}$$

Assuming that $\mathbf{I}_{n_\ell} - q_\ell \hat{q}_{\ell+1} \tilde{\boldsymbol{\Sigma}}_\ell$ is invertible, we may use the multiplicative property of the determinant and the mixed-product property of the Kronecker product to expand this as

$$\det(\mathbf{I}_{n_\ell} - q_\ell \hat{q}_{\ell+1} \tilde{\boldsymbol{\Sigma}}_\ell)^m$$
$$\times \det\{\mathbf{I}_{mn_\ell} - (q_\ell \hat{c}_{\ell+1} + c_\ell \hat{q}_{\ell+1} + mc_\ell \hat{c}_{\ell+1})[\mathbf{1}_m \otimes (\mathbf{I}_{n_\ell} - q_\ell \hat{q}_{\ell+1} \tilde{\boldsymbol{\Sigma}}_\ell)^{-1} \boldsymbol{\Sigma}_\ell](\mathbf{1}_m^\top \otimes \mathbf{I}_{n_\ell})\}. \tag{154}$$

Then, by the Weinstein–Aronszajn identity, we have

$$\det\{\mathbf{I}_{mn_\ell} - (q_\ell \hat{c}_{\ell+1} + c_\ell \hat{q}_{\ell+1} + mc_\ell \hat{c}_{\ell+1})[\mathbf{1}_m \otimes (\mathbf{I}_{n_\ell} - q_\ell \hat{q}_{\ell+1} \tilde{\boldsymbol{\Sigma}}_\ell)^{-1} \boldsymbol{\Sigma}_\ell](\mathbf{1}_m^\top \otimes \mathbf{I}_{n_\ell})\} \tag{155}$$
$$= \det\{\mathbf{I}_{n_\ell} - (q_\ell \hat{c}_{\ell+1} + c_\ell \hat{q}_{\ell+1} + mc_\ell \hat{c}_{\ell+1})(\mathbf{1}_m^\top \otimes \mathbf{I}_{n_\ell})[\mathbf{1}_m \otimes (\mathbf{I}_{n_\ell} - q_\ell \hat{q}_{\ell+1} \tilde{\boldsymbol{\Sigma}}_\ell)^{-1} \boldsymbol{\Sigma}_\ell]\} \tag{156}$$
$$= \det[\mathbf{I}_{n_\ell} - m(q_\ell \hat{c}_{\ell+1} + c_\ell \hat{q}_{\ell+1} + mc_\ell \hat{c}_{\ell+1})(\mathbf{I}_{n_\ell} - q_\ell \hat{q}_{\ell+1} \tilde{\boldsymbol{\Sigma}}_\ell)^{-1} \boldsymbol{\Sigma}_\ell]. \tag{157}$$

This yields

$$\frac{1}{mn_\ell} \det[\mathbf{I}_{mn_\ell} - (\mathbf{C}_\ell \hat{\mathbf{C}}_{\ell+1}) \otimes \tilde{\boldsymbol{\Sigma}}_\ell] \tag{158}$$
$$= \frac{1}{n_\ell} \log \det(\mathbf{I}_{n_\ell} - q_\ell \hat{q}_{\ell+1} \tilde{\boldsymbol{\Sigma}}_\ell)$$
$$+ \frac{1}{mn_\ell} \log \det[\mathbf{I}_{n_\ell} - m(q_\ell \hat{c}_{\ell+1} + c_\ell \hat{q}_{\ell+1} + mc_\ell \hat{c}_{\ell+1})(\mathbf{I}_{n_\ell} - q_\ell \hat{q}_{\ell+1} \tilde{\boldsymbol{\Sigma}}_\ell)^{-1} \boldsymbol{\Sigma}_\ell]. \tag{159}$$

Here, we write $\mathbb{E}_{\tilde{\sigma}_\ell}$ for expectation with respect to the limiting empirical distribution of eigenvalues of the matrix $\tilde{\boldsymbol{\Sigma}}_\ell$. Assuming no issues arise in interchanging limits in $m$ and $n_\ell$, we can then use the series expansion of the log-determinant near the identity [33] to obtain

$$\frac{1}{mn_\ell} \log \det[\mathbf{I}_{n_\ell} - m(q_\ell \hat{c}_{\ell+1} + c_\ell \hat{q}_{\ell+1} + mc_\ell \hat{c}_{\ell+1})(\mathbf{I}_{n_\ell} - q_\ell \hat{q}_{\ell+1} \tilde{\boldsymbol{\Sigma}}_\ell)^{-1} \boldsymbol{\Sigma}_\ell] \tag{160}$$
$$= -(q_\ell \hat{c}_{\ell+1} + c_\ell \hat{q}_{\ell+1}) \frac{1}{n_\ell} \operatorname{tr}[(\mathbf{I}_{n_\ell} - q_\ell \hat{q}_{\ell+1} \tilde{\boldsymbol{\Sigma}}_\ell)^{-1} \boldsymbol{\Sigma}_\ell] + \mathcal{O}(m) \tag{161}$$
$$= -(q_\ell \hat{c}_{\ell+1} + c_\ell \hat{q}_{\ell+1}) \mathbb{E}_{\tilde{\sigma}_\ell} \left[ \frac{\tilde{\sigma}_\ell}{1 - q_\ell \hat{q}_{\ell+1} \tilde{\sigma}_\ell} \right] + \mathcal{O}(m). \tag{162}$$

Therefore, we have

$$\frac{1}{mn_\ell} \det[\mathbf{I}_{mn_\ell} - (\mathbf{C}_\ell \hat{\mathbf{C}}_{\ell+1}) \otimes \tilde{\boldsymbol{\Sigma}}_\ell] = \mathbb{E}_{\tilde{\sigma}_\ell} \log(1 - q_\ell \hat{q}_{\ell+1} \tilde{\sigma}_\ell)$$
$$- (q_\ell \hat{c}_{\ell+1} + c_\ell \hat{q}_{\ell+1}) \mathbb{E}_{\tilde{\sigma}_\ell} \left[ \frac{\tilde{\sigma}_\ell}{1 - q_\ell \hat{q}_{\ell+1} \tilde{\sigma}_\ell} \right]$$
$$+ \mathcal{O}(m). \tag{163}$$

By an identical argument, we have

$$\frac{1}{mn_0} \det(\hat{\mathbf{C}}_1 \otimes \boldsymbol{\Gamma}_1 - z\mathbf{I}_{mn_0}) = \mathbb{E}_{\gamma_1} \log(\gamma_1 \hat{q}_1 - z) + \mathbb{E}_{\gamma_1} \left[ \frac{\gamma_1 \hat{c}_1}{\gamma_1 \hat{q}_1 - z} \right] + \mathcal{O}(m). \tag{164}$$

Combining these results, we obtain

$$\lim_{m \to 0} S = \mathbb{E}_{\gamma_1} \log(\gamma_1 \hat{q}_1 - z) + \mathbb{E}_{\gamma_1} \left[ \frac{\gamma_1 \hat{c}_1}{\gamma_1 \hat{q}_1 - z} \right] - \log(2\pi)$$
$$+ \sum_{\ell=1}^{L} \alpha_\ell \left( q_\ell \hat{q}_\ell + q_\ell \hat{c}_\ell + c_\ell \hat{q}_\ell + \mathbb{E}_{\sigma_\ell} \log(1 - q_\ell \hat{q}_{\ell+1} \sigma_\ell) \right.$$
$$\left. - (q_\ell \hat{c}_{\ell+1} + c_\ell \hat{q}_{\ell+1}) \mathbb{E}_{\sigma_\ell} \left[ \frac{\sigma_\ell}{1 - q_\ell \hat{q}_{\ell+1} \sigma_\ell} \right] \right) \tag{165}$$

with the boundary condition $\hat{q}_{L+1} = 1$, $\hat{c}_{L+1} = 0$.

Moreover, we have

$$G(z) = -\lim_{m \to 0} \frac{\partial S}{\partial z} = -\mathbb{E}_{\gamma_1} \left[ \frac{1}{z - \gamma_1 \hat{q}_1} \right] - \mathbb{E}_{\gamma_1} \left[ \frac{\gamma_1 \hat{c}_1}{(z - \gamma_1 \hat{q}_1)^2} \right], \tag{166}$$

where the order parameters are to be evaluated at their saddle point values.

From the equations $\partial S / \partial q_\ell = 0$, we have

$$0 = \hat{q}_\ell + \hat{c}_\ell - (\hat{q}_{\ell+1} + \hat{c}_{\ell+1}) \mathbb{E}_{\tilde{\sigma}_\ell} \left[ \frac{\tilde{\sigma}_\ell}{1 - q_\ell \hat{q}_{\ell+1} \tilde{\sigma}_\ell} \right] - (q_\ell \hat{c}_{\ell+1} + c_\ell \hat{q}_{\ell+1}) \hat{q}_{\ell+1} \mathbb{E}_{\tilde{\sigma}_\ell} \left[ \left( \frac{\tilde{\sigma}_\ell}{1 - q_\ell \hat{q}_{\ell+1} \tilde{\sigma}_\ell} \right)^2 \right] \tag{167}$$

for all $\ell = 1, \dots, L$. From the equations $\partial S / \partial \hat{q}_\ell = 0$, we have

$$0 = -\mathbb{E}_{\gamma_1} \left[ \frac{\gamma_1}{z - \gamma_1 \hat{q}_1} \right] - \mathbb{E}_{\gamma_1} \left[ \frac{\gamma_1^2 \hat{c}_1}{(z - \gamma_1 \hat{q}_1)^2} \right] + \alpha_1 (q_1 + c_1) \tag{168}$$

for $\ell = 1$ and

$$0 = \alpha_\ell (q_\ell + c_\ell) - \alpha_{\ell-1} (q_{\ell-1} + c_{\ell-1}) \mathbb{E}_{\tilde{\sigma}_{\ell-1}} \left[ \frac{\tilde{\sigma}_{\ell-1}}{1 - q_{\ell-1} \hat{q}_\ell \tilde{\sigma}_{\ell-1}} \right]$$
$$- \alpha_{\ell-1} (q_{\ell-1} \hat{c}_\ell + c_{\ell-1} \hat{q}_\ell) q_{\ell-1} \mathbb{E}_{\tilde{\sigma}_{\ell-1}} \left[ \left( \frac{\tilde{\sigma}_{\ell-1}}{1 - q_{\ell-1} \hat{q}_\ell \tilde{\sigma}_{\ell-1}} \right)^2 \right] \tag{169}$$

for $\ell = 2, \dots, L$. From $\partial S / \partial c_\ell = 0$, we have

$$0 = \hat{q}_\ell - \hat{q}_{\ell+1} \mathbb{E}_{\tilde{\sigma}_\ell} \left[ \frac{\tilde{\sigma}_\ell}{1 - q_\ell \hat{q}_{\ell+1} \tilde{\sigma}_\ell} \right] \tag{170}$$

for $\ell = 1, \dots, L$. Finally, from $\partial S / \partial \hat{c}_\ell = 0$, we have

$$0 = -\mathbb{E}_{\gamma_1} \left[ \frac{\gamma_1}{z - \gamma_1 \hat{q}_1} \right] + \alpha_1 q_1 \tag{171}$$

for $\ell = 1$ and

$$0 = \alpha_\ell q_\ell - \alpha_{\ell-1} q_{\ell-1} \mathbb{E}_{\tilde{\sigma}_{\ell-1}} \left[ \frac{\tilde{\sigma}_{\ell-1}}{1 - q_{\ell-1} \hat{q}_\ell \tilde{\sigma}_{\ell-1}} \right] \tag{172}$$

for $\ell = 2, \dots, L$.

As in the unstructured case, we can decouple the replica-uniform components from the replica-uniform components. This yields the system

$$\hat{q}_\ell = \hat{q}_{\ell+1} \mathbb{E}_{\tilde{\sigma}_\ell} \left[ \frac{\tilde{\sigma}_\ell}{1 - q_\ell \hat{q}_{\ell+1} \tilde{\sigma}_\ell} \right] \qquad (\ell = 1, \dots, L) \tag{173}$$

$$q_1 = \frac{1}{\alpha_1} \mathbb{E}_{\gamma_1} \left[ \frac{\gamma_1}{z - \gamma_1 \hat{q}_1} \right] \tag{174}$$

$$q_\ell = \frac{\alpha_{\ell-1}}{\alpha_\ell} q_{\ell-1} \mathbb{E}_{\tilde{\sigma}_{\ell-1}} \left[ \frac{\tilde{\sigma}_{\ell-1}}{1 - q_{\ell-1} \hat{q}_\ell \tilde{\sigma}_{\ell-1}} \right] \qquad (\ell = 2, \dots, L) \tag{175}$$

for the non-uniform components. Given a solution to that system, the replica-uniform components are determined by the linear system

$$\hat{c}_\ell = \hat{c}_{\ell+1} \mathbb{E}_{\tilde{\sigma}_\ell} \left[ \frac{\tilde{\sigma}_\ell}{1 - q_\ell \hat{q}_{\ell+1} \tilde{\sigma}_\ell} \right]$$

$$+ (q_\ell \hat{c}_{\ell+1} + c_\ell \hat{q}_{\ell+1}) \hat{q}_{\ell+1} \mathbb{E}_{\tilde{\sigma}_\ell} \left[ \left( \frac{\tilde{\sigma}_\ell}{1 - q_\ell \hat{q}_{\ell+1} \tilde{\sigma}_\ell} \right)^2 \right] \qquad (\ell = 1, \dots, L) \qquad (176)$$

$$c_1 = \frac{1}{\alpha_1} \mathbb{E}_{\gamma_1} \left[ \frac{\gamma_1^2 \hat{c}_1}{(z - \gamma_1 \hat{q}_1)^2} \right] \qquad (177)$$

$$c_\ell = \frac{\alpha_{\ell-1}}{\alpha_\ell} c_{\ell-1} \mathbb{E}_{\tilde{\sigma}_{\ell-1}} \left[ \frac{\tilde{\sigma}_{\ell-1}}{1 - q_{\ell-1} \hat{q}_\ell \tilde{\sigma}_{\ell-1}} \right]$$

$$+ \frac{\alpha_{\ell-1}}{\alpha_\ell} (q_{\ell-1} \hat{c}_\ell + c_{\ell-1} \hat{q}_\ell) q_{\ell-1} \mathbb{E}_{\tilde{\sigma}_{\ell-1}} \left[ \left( \frac{\tilde{\sigma}_{\ell-1}}{1 - q_{\ell-1} \hat{q}_\ell \tilde{\sigma}_{\ell-1}} \right)^2 \right] \qquad (\ell = 2, \dots, L). \qquad (178)$$

Recalling the boundary condition $\hat{q}_{L+1} = 1$, $\hat{c}_{L+1} = 0$, it is easy to see that we should have $c_\ell = \hat{c}_\ell = 0$ for all $\ell = 1, \dots, L$. Thus, as in the unstructured case, the annealed average is exact.

Our task is therefore to solve the system of equations for the replica non-uniform components of the order parameters,

$$\hat{q}_\ell = \hat{q}_{\ell+1} \mathbb{E}_{\tilde{\sigma}_\ell} \left[ \frac{\tilde{\sigma}_\ell}{1 - q_\ell \hat{q}_{\ell+1} \tilde{\sigma}_\ell} \right] \qquad (\ell = 1, \dots, L) \qquad (179)$$

$$q_1 = \frac{1}{\alpha_1} \mathbb{E}_{\gamma_1} \left[ \frac{\gamma_1}{z - \gamma_1 \hat{q}_1} \right] \qquad (180)$$

$$q_\ell = \frac{\alpha_{\ell-1}}{\alpha_\ell} q_{\ell-1} \mathbb{E}_{\tilde{\sigma}_{\ell-1}} \left[ \frac{\tilde{\sigma}_{\ell-1}}{1 - q_{\ell-1} \hat{q}_\ell \tilde{\sigma}_{\ell-1}} \right] \qquad (\ell = 2, \dots, L), \qquad (181)$$

subject to the boundary condition $\hat{q}_{L+1} = 1$, in terms of which the resolvent is given as

$$G(z) = -\mathbb{E}_{\gamma_1} \left[ \frac{1}{z - \gamma_1 \hat{q}_1} \right]. \qquad (182)$$

As a sanity check, we can see immediately that this reduces to our earlier result in the unstructured case $\Sigma_\ell = \mathbf{I}_{n_\ell}$, $\Gamma_\ell = \mathbf{I}_{n_{\ell-1}}$.

We start by writing these equations in terms of standard objects in random matrix theory. We have

$$\mathbb{E}_{\tilde{\sigma}_\ell} \left[ \frac{q_\ell \hat{q}_{\ell+1} \tilde{\sigma}_\ell}{1 - q_\ell \hat{q}_{\ell+1} \tilde{\sigma}_\ell} \right] = M_{\tilde{\Sigma}_\ell} \left( \frac{1}{q_\ell \hat{q}_{\ell+1}} \right) \qquad (183)$$

for $M_{\tilde{\Sigma}_\ell}(z)$ the moment generating function of $\tilde{\Sigma}_\ell$. Similarly, we have

$$\mathbb{E}_{\gamma_1} \left[ \frac{\hat{q}_1 \gamma_1}{z - \gamma_1 \hat{q}_1} \right] = M_{\Gamma_1} \left( \frac{z}{\hat{q}_1} \right). \qquad (184)$$

Then, we have

$$q_\ell \hat{q}_\ell = M_{\tilde{\Sigma}_\ell} \left( \frac{1}{q_\ell \hat{q}_{\ell+1}} \right) \qquad (\ell = 1, \dots, L) \qquad (185)$$

$$q_1 \hat{q}_1 = \frac{1}{\alpha_1} M_{\Gamma_1} \left( \frac{z}{\hat{q}_1} \right) \qquad (186)$$

$$q_\ell \hat{q}_\ell = \frac{\alpha_{\ell-1}}{\alpha_\ell} M_{\tilde{\Sigma}_{\ell-1}} \left( \frac{1}{q_{\ell-1} \hat{q}_\ell} \right) \qquad (\ell = 2, \dots, L). \qquad (187)$$

Moreover, we observe that the equation for $G(z)$ implies that

$$\alpha_1 q_1 \hat{q}_1 = \mathbb{E}_{\gamma_1}\left[\frac{\gamma_1 \hat{q}_1}{z - \gamma_1 \hat{q}_1}\right] \tag{188}$$

$$= \mathbb{E}_{\gamma_1}\left[\frac{z}{z - \gamma_1 \hat{q}_1}\right] - 1 \tag{189}$$

$$= -zG(z) - 1 \tag{190}$$

$$= M(z). \tag{191}$$

Thus, for $\ell = 2, \ldots, L$, we have

$$q_\ell \hat{q}_\ell = \frac{\alpha_{\ell-1}}{\alpha_\ell} M_{\tilde{\Sigma}_{\ell-1}}\left(\frac{1}{q_{\ell-1}\hat{q}_\ell}\right) \tag{192}$$

$$= \frac{\alpha_{\ell-1}}{\alpha_\ell} q_{\ell-1}\hat{q}_{\ell-1}. \tag{193}$$

This relation can easily be iterated backward to give

$$q_\ell \hat{q}_\ell = \frac{\alpha_1}{\alpha_\ell} q_1 \hat{q}_1. \tag{194}$$

for all $\ell = 1, \ldots, L$, where the $\ell = 1$ case is of course a tautology. But, we have $\alpha_1 q_1 \hat{q}_1 = M(z)$, hence we obtain

$$M(z) = \alpha_\ell q_\ell \hat{q}_\ell = \alpha_\ell M_{\tilde{\Sigma}_\ell}\left(\frac{1}{q_\ell \hat{q}_{\ell+1}}\right) \tag{195}$$

for $\ell = 1, \ldots, L$. Assuming the invertibility of $M_{\tilde{\Sigma}_\ell}$, we therefore have

$$\frac{1}{q_\ell \hat{q}_{\ell+1}} = M_{\tilde{\Sigma}_\ell}^{-1}\left(\frac{M(z)}{\alpha_\ell}\right) \tag{196}$$

for $\ell = 1, \ldots, L$. Using the boundary condition $\hat{q}_{L+1} = 1$, we have

$$\frac{1}{q_L} = M_{\tilde{\Sigma}_L}^{-1}\left(\frac{M(z)}{\alpha_L}\right). \tag{197}$$

For $\ell = 1, \ldots, L-1$, we can multiply through by $q_{\ell+1}\hat{q}_{\ell+1}$ to obtain

$$\frac{q_{\ell+1}}{q_\ell} = \frac{M(z)}{\alpha_{\ell+1}} M_{\tilde{\Sigma}_\ell}^{-1}\left(\frac{M(z)}{\alpha_\ell}\right). \tag{198}$$

This gives

$$\frac{1}{q_\ell} = \frac{1}{q_L} \prod_{\ell=j}^{L-1} \frac{q_{j+1}}{q_j} \tag{199}$$

$$= M_{\tilde{\Sigma}_L}^{-1}\left(\frac{M(z)}{\alpha_L}\right) \prod_{j=\ell}^{L-1}\left[\frac{M(z)}{\alpha_{j+1}} M_{\tilde{\Sigma}_j}^{-1}\left(\frac{M(z)}{\alpha_j}\right)\right] \tag{200}$$

$$= \frac{\alpha_\ell}{M(z)} \prod_{j=\ell}^{L}\left[\frac{M(z)}{\alpha_j} M_{\tilde{\Sigma}_j}^{-1}\left(\frac{M(z)}{\alpha_j}\right)\right]. \tag{201}$$

Using the relation $\alpha_\ell q_\ell \hat{q}_\ell = M(z)$, we have

$$\hat{q}_\ell = \prod_{j=\ell}^{L} \left[ \frac{M(z)}{\alpha_\ell} M_{\tilde{\Sigma}_j}^{-1} \left( \frac{M(z)}{\alpha_\ell} \right) \right]. \tag{202}$$

We can now finally use the equation

$$M(z) = \alpha_1 q_1 \hat{q}_1 = M_{\Gamma_1} \left( \frac{z}{\hat{q}_1} \right) \tag{203}$$

to write

$$\hat{q}_1 = \frac{z}{M_{\Gamma_1}^{-1}(M(z))}, \tag{204}$$

hence we obtain the closed equation

$$\frac{z}{M_{\Gamma_1}^{-1}(M(z))} = \prod_{\ell=1}^{L} \left[ \frac{M(z)}{\alpha_\ell} M_{\tilde{\Sigma}_\ell}^{-1} \left( \frac{M(z)}{\alpha_\ell} \right) \right]. \tag{205}$$

This is the result claimed in (27).

## C  Computing the extremal eigenvalues for unstructured factors

In this appendix, we use the method outlined in §3.1 to obtain the conditions reported in §3.2 on the maximum and minimum eigenvalues of Wishart product matrices with unstructured factors. As in our derivation of the Stieljes transform in §A, we divide the replica computation of the minimum and maximum eigenvalues into two parts. We first compute the moments of the partition function in §C.1, and then simplify the replica-symmetric saddle point equations in §C.2.

### C.1  Step I: Evaluating the moments of the partition function

Again, we introduce replicas indexed by $a = 1, \ldots, m$, which gives the moments of the partition function for the spherical spin glass (42) as

$$\mathbb{E}Z^m = \int \prod_a d\mathbf{w}^a \left[ \prod_{a=1}^{m} \delta \left( 1 - \frac{1}{n_0} \|\mathbf{w}^a\|^2 \right) \right] \mathbb{E} \exp \left( \frac{-\beta}{2 n_L \cdots n_1} \sum_{a=1}^{m} (\mathbf{w}^a)^\top \mathbf{X}_1^\top \cdots \mathbf{X}_L^\top \mathbf{X}_L \cdots \mathbf{X}_1 \mathbf{w}^a \right), \tag{206}$$

where we enforce the spherical constraints with $\delta$-distributions. It is easy to see that the matrices $\mathbf{X}_\ell$ can be integrated out much as before, except for the fact that the order parameters we introduce should be real, i.e.,

$$C_\ell^{ab} \equiv \frac{1}{n_1 \cdots n_\ell} (\mathbf{w}^a)^\top \mathbf{X}_1^\top \cdots \mathbf{X}_{\ell-1}^\top \mathbf{X}_{\ell-1} \cdots \mathbf{X}_1 \mathbf{w}^b, \tag{207}$$

and that the boundary condition is now $\hat{\mathbf{C}}_{L+1} = -\beta \mathbf{I}_m$. Iterating backwards, this yields

$$\mathbb{E} Z^m = \int \frac{d\mathbf{C}_2 \, d\hat{\mathbf{C}}_2}{(4\pi i/n_2)^{m(m+1)/2}} \cdots \int \frac{d\mathbf{C}_L \, d\hat{\mathbf{C}}_L}{(4\pi i/n_L)^{m(m+1)/2}}$$
$$\exp\left(-\frac{1}{2} \sum_{\ell=2}^{L} n_\ell [\mathrm{tr}(\mathbf{C}_\ell \hat{\mathbf{C}}_\ell) + \log \det(\mathbf{I}_m - \mathbf{C}_\ell \hat{\mathbf{C}}_{\ell+1})]\right)$$
$$\times \int \prod_{a=1}^{m} d\mathbf{w}^a \left[\prod_{a=1}^{m} \delta\left(1 - \frac{1}{n_0}\|\mathbf{w}^a\|^2\right)\right] \det(\mathbf{I}_m - \mathbf{C}_1 \hat{\mathbf{C}}_2)^{-n_1/2}, \tag{208}$$

where we recall that

$$C_1^{ab} = \frac{1}{n_1}(\mathbf{w}^a)^\top \mathbf{w}^b. \tag{209}$$

By the spherical constraint, we have

$$C_1^{aa} = \frac{n_0}{n_1} = \frac{1}{\alpha_1}. \tag{210}$$

It is therefore useful to instead introduce order parameters

$$F^{ab} = \frac{1}{n_0}(\mathbf{w}^a)^\top \mathbf{w}^b \tag{211}$$

via Fourier representations of the $\delta$-distribution, such that $F^{aa} = 1$ and $\mathbf{C}_1 = \mathbf{F}/\alpha_1$. Integrating over $\mathbf{F}$ with $F^{aa} = 1$, the corresponding Lagrange multipliers $\hat{F}^{aa}$ automatically enforce the spherical constraint. Then, after evaluating the remaining unconstrained Gaussian integral over $\mathbf{w}^a$, we obtain

$$\mathbb{E} Z^m \propto \int \frac{d\mathbf{F} \, d\hat{\mathbf{F}}}{(4\pi i/n_0)^{m(m+1)/2}} \int \frac{d\mathbf{C}_2 \, d\hat{\mathbf{C}}_2}{(4\pi i/n_2)^{m(m+1)/2}} \cdots \int \frac{d\mathbf{C}_L \, d\hat{\mathbf{C}}_L}{(4\pi i/n_L)^{m(m+1)/2}} \exp\left(\frac{n_0 m}{2} S\right) \tag{212}$$

for

$$S(\mathbf{F}, \hat{\mathbf{F}}, \mathbf{C}_2, \hat{\mathbf{C}}_2, \cdots, \mathbf{C}_L, \hat{\mathbf{C}}_L) = \frac{1}{m} \mathrm{tr}(\mathbf{F}\hat{\mathbf{F}}) - \frac{1}{m} \log \det(\hat{\mathbf{F}}) - \frac{1}{m} \alpha_1 \log \det(\mathbf{I}_m - \alpha_1^{-1} \mathbf{F} \hat{\mathbf{C}}_2)$$
$$- \frac{1}{m} \sum_{\ell=2}^{L} \alpha_\ell [\mathrm{tr}(\mathbf{C}_\ell \hat{\mathbf{C}}_\ell) + \log \det(\mathbf{I}_m - \mathbf{C}_\ell \hat{\mathbf{C}}_{\ell+1})]). \tag{213}$$

As in our computation of the Stieltjes transform, this integral can be evaluated using the method of steepest descent, yielding

$$g = - \mathop{\mathrm{extr}}_{\mathbf{C}_1, \hat{\mathbf{C}}_1, \dots, \mathbf{C}_L, \hat{\mathbf{C}}_L} S. \tag{214}$$

Again, we will consider only replica-symmetric saddle points.

## C.2  Step II: The replica-symmetric saddle point equations

We make an RS *Ansatz*

$$\mathbf{F} = (1-f)\mathbf{I}_m + f\,\mathbf{1}_m\mathbf{1}_m^\top \tag{215}$$

$$\hat{\mathbf{F}} = (\hat{F}-\hat{f})\mathbf{I}_m + \hat{f}\,\mathbf{1}_m\mathbf{1}_m^\top \tag{216}$$

$$\mathbf{C}_\ell = q_\ell\mathbf{I}_m + c_\ell\,\mathbf{1}_m\mathbf{1}_m^\top \qquad\qquad (\ell = 2,\ldots,L) \tag{217}$$

$$\hat{\mathbf{C}}_\ell = \hat{q}_\ell\mathbf{I}_m + \hat{c}_\ell\,\mathbf{1}_m\mathbf{1}_m^\top \qquad\qquad (\ell = 2,\ldots,L). \tag{218}$$

Again, we use standard identities to obtain

$$\lim_{m\to 0}\frac{1}{m}\log\det(\hat{\mathbf{F}}) = \log(\hat{F}-\hat{f}) + \frac{\hat{f}}{\hat{F}-\hat{f}}, \tag{219}$$

$$\lim_{m\to 0}\frac{1}{m}\mathrm{tr}(\mathbf{F}\hat{\mathbf{F}}) = \hat{F} - f\hat{f}, \tag{220}$$

and

$$\lim_{m\to 0}\frac{1}{m}\log\det(\mathbf{I}_m - \alpha_1^{-1}\mathbf{F}\hat{\mathbf{C}}_2) = \log(1-\alpha_1^{-1}(1-f)\hat{q}_2) - \frac{\alpha_1^{-1}(1-f)\hat{c}_2 + \alpha_1^{-1}f\hat{q}_2}{1-\alpha_1^{-1}(1-f)\hat{q}_2}, \tag{221}$$

yielding

$$\lim_{m\to 0}S = \hat{F} - f\hat{f} - \log(\hat{F}-\hat{f}) - \frac{\hat{f}}{\hat{F}-\hat{f}}$$
$$- \alpha_1\left(\log(1-\alpha_1^{-1}(1-f)\hat{q}_2) - \frac{\alpha_1^{-1}(1-f)\hat{c}_2 + \alpha_1^{-1}f\hat{q}_2}{1-\alpha_1^{-1}(1-f)\hat{q}_2}\right)$$
$$- \sum_{\ell=2}^{L}\alpha_\ell\left(q_\ell\hat{q}_\ell + q_\ell\hat{c}_\ell + c_\ell\hat{q}_\ell + \log(1-q_\ell\hat{q}_{\ell+1}) - \frac{q_\ell\hat{c}_{\ell+1} + c_\ell\hat{q}_{\ell+1}}{1-q_\ell\hat{q}_{\ell+1}}\right), \tag{222}$$

where we recall the endpoint condition $\hat{q}_{L+1} = -\beta$, $\hat{c}_{L+1} = 0$.

For brevity, we define $q_1 = \alpha_1^{-1}(1-f)$ and $c_1 = \alpha_1^{-1}f$. Then, by comparison with our previous results, the saddle point equations for $\ell = 2,\ldots,L$ are

$$\hat{q}_\ell = \frac{\hat{q}_{\ell+1}}{1-q_\ell\hat{q}_{\ell+1}} \tag{223}$$

$$q_\ell = \frac{\alpha_{\ell-1}}{\alpha_\ell}\frac{q_{\ell-1}}{1-q_{\ell-1}\hat{q}_\ell} \tag{224}$$

$$\hat{c}_\ell = \frac{\hat{c}_{\ell+1} + c_\ell\hat{q}_{\ell+1}^2}{(1-q_\ell\hat{q}_{\ell+1})^2} \tag{225}$$

$$c_\ell = \frac{\alpha_{\ell-1}}{\alpha_\ell}\frac{c_{\ell-1} + q_{\ell-1}^2\hat{c}_\ell}{(1-q_{\ell-1}\hat{q}_\ell)^2}. \tag{226}$$

The saddle point equation $\partial S/\partial\hat{F} = 0$ yields

$$0 = 1 + \frac{\hat{f}}{(\hat{F}-\hat{f})^2} - \frac{1}{\hat{F}-\hat{f}}, \tag{227}$$

while the equation $\partial S/\partial \hat{f} = 0$ yields

$$0 = -f - \frac{\hat{f}}{(\hat{F} - \hat{f})^2}, \tag{228}$$

hence we have

$$\hat{F} - \hat{f} = \frac{1}{1 - f} \tag{229}$$

and

$$\hat{f} = -\frac{f}{(1-f)^2}. \tag{230}$$

Finally, the equation $\partial S/\partial f = 0$ yields

$$0 = -\hat{f} - \frac{\hat{q}_2^2 c_1 + \hat{c}_2}{(1 - q_1 \hat{q}_2)^2} \tag{231}$$

Then, we can easily eliminate the Lagrange multipliers $\hat{F}$ and $\hat{f}$. The remaining system can be written compactly as

$$\hat{q}_\ell = \frac{\hat{q}_{\ell+1}}{1 - q_\ell \hat{q}_{\ell+1}} \qquad (\ell = 2, \ldots, L) \tag{232}$$

$$q_\ell = \frac{\alpha_{\ell-1}}{\alpha_\ell} \frac{q_{\ell-1}}{1 - q_{\ell-1}\hat{q}_\ell} \qquad (\ell = 2, \ldots, L) \tag{233}$$

$$\hat{c}_\ell = \frac{\hat{c}_{\ell+1} + c_\ell \hat{q}_{\ell+1}^2}{(1 - q_\ell \hat{q}_{\ell+1})^2} \qquad (\ell = 1, \ldots, L) \tag{234}$$

$$c_\ell = \frac{\alpha_{\ell-1}}{\alpha_\ell} \frac{c_{\ell-1} + q_{\ell-1}^2 \hat{c}_\ell}{(1 - q_{\ell-1}\hat{q}_\ell)^2} \qquad (\ell = 2, \ldots, L), \tag{235}$$

where we have the definitions

$$q_1 \equiv \alpha_1^{-1}(1 - f) \tag{236}$$

$$c_1 \equiv \alpha_1^{-1} f \tag{237}$$

$$\hat{c}_1 \equiv \frac{f}{(1-f)^2} \tag{238}$$

and the endpoint conditions

$$\hat{q}_{L+1} = -\beta \tag{239}$$

$$\hat{c}_{L+1} = 0. \tag{240}$$

Moreover, we have

$$\mathbb{E}\lambda_{\min} = -\lim_{\beta \to \infty} \lim_{m \to 0} \frac{\partial S}{\partial \beta} \tag{241}$$

$$= \lim_{\beta \to \infty} \lim_{m \to 0} \frac{\partial S}{\partial \hat{q}_{L+1}} \tag{242}$$

$$= -\alpha_L \lim_{\beta \to \infty} \frac{\partial}{\partial \hat{q}_{L+1}} \left( \log(1 - q_L \hat{q}_{L+1}) - \frac{c_L \hat{q}_{L+1}}{1 - q_L \hat{q}_{L+1}} \right) \tag{243}$$

$$= \alpha_L \lim_{\beta \to \infty} \left( \frac{q_L}{1 + \beta q_L} + \frac{c_L}{(1 + \beta q_L)^2} \right), \tag{244}$$

where the order parameters are to be evaluated at their saddle point values. Our task is therefore to solve the saddle point equations in the zero temperature limit.

We first simplify the replica-nonuniform saddle point equations using the same trick as before. To do so, it is useful to define an auxiliary variable $\hat{q}_1$ by

$$\hat{q}_1 = \frac{\hat{q}_2}{1 - q_1 \hat{q}_2}, \tag{245}$$

such that the system of equations is identical to what we encountered in §A.2. Then, letting

$$A = \alpha_1 q_1 \hat{q}_1, \tag{246}$$

we have the backward recurrence

$$\hat{q}_\ell = \left(1 + \frac{A}{\alpha_\ell}\right) \hat{q}_{\ell+1} \tag{247}$$

for $\ell = 1, \ldots, L$, which can be solved using the endpoint condition $\hat{q}_{L+1} = -\beta$, yielding

$$\hat{q}_\ell = -\beta \prod_{j=\ell}^{L} \left(1 + \frac{A}{\alpha_j}\right). \tag{248}$$

This shows that we should have $\hat{q}_\ell \sim \mathcal{O}(\beta)$ and $q_\ell \sim \mathcal{O}(1/\beta)$. With these scalings, we have

$$\mathbb{E}\lambda_{\min} = \alpha_L \lim_{\beta\to\infty} \frac{c_L}{(1 + \beta q_L)^2}. \tag{249}$$

To obtain $q_L$, we use the equation

$$q_\ell = \frac{A}{\alpha_\ell \hat{q}_\ell} \tag{250}$$

which gives

$$q_L = \frac{A}{\alpha_L \hat{q}_L} = -\frac{A}{\beta(\alpha_L + A)}, \tag{251}$$

hence

$$\mathbb{E}\lambda_{\min} = \alpha_L \lim_{\beta\to\infty} \left(\frac{\alpha_L + A}{\alpha_L}\right)^2 c_L. \tag{252}$$

The equations for the replica-uniform components can be simplified after a bit of tedious but straightforward algebra. Deferring the details of this computation to Appendix C.3, we obtain an expression for $c_L$ in terms of $c_1$,

$$c_L = \frac{\alpha_1 c_1}{\alpha_L} \frac{\hat{q}_1^2}{\hat{q}_L^2} \left(\prod_{j=1}^{L} \frac{\alpha_j}{\alpha_j + A}\right) \frac{1}{1 - \sum_{j=1}^{L} \frac{A}{a_j + A}}. \tag{253}$$

along with the condition

$$\hat{c}_1 = \frac{\hat{q}_1}{q_1} c_1 \frac{\sum_{j=1}^{L} \frac{A}{\alpha_j + A}}{1 - \sum_{j=1}^{L} \frac{A}{a_j + A}}, \tag{254}$$

where we again have defined $A = \alpha_1 q_1 \hat{q}_1$. Recalling the definitions

$$q_1 \equiv \alpha_1^{-1}(1-f) \tag{255}$$

$$c_1 \equiv \alpha_1^{-1} f \tag{256}$$

$$\hat{c}_1 \equiv \frac{f}{(1-f)^2}, \tag{257}$$

we can use the condition on $\hat{c}_1$ to obtain a closed equation for $A$,

$$\frac{1}{A} = \frac{\sum_{j=1}^{L} \frac{A}{\alpha_j + A}}{1 - \sum_{j=1}^{L} \frac{A}{\alpha_j + A}}. \tag{258}$$

Then, recalling that

$$\hat{q}_\ell = -\beta \prod_{j=\ell}^{L} \left(1 + \frac{A}{\alpha_j}\right), \tag{259}$$

we have

$$\frac{\hat{q}_1^2}{\hat{q}_L^2} = \prod_{j=1}^{L-1} \left(\frac{\alpha_j + A}{\alpha_j}\right)^2 \tag{260}$$

so

$$\mathbb{E}\lambda_{\min} = \lim_{\beta \to \infty} \alpha_L \left(\frac{\alpha_L + A}{\alpha_L}\right)^2 c_L \tag{261}$$

$$= \lim_{\beta \to \infty} f \frac{1}{1 - \sum_{j=1}^{L} \frac{A}{\alpha_j + A}} \prod_{\ell=1}^{L} \frac{\alpha_\ell + A}{\alpha_\ell}. \tag{262}$$

To solve these equations in the limit $\beta \to \infty$, it is clear that we should have $\hat{q}_1 \sim \mathcal{O}(\beta)$ and $1 - f \sim \mathcal{O}(1/\beta)$, such that $A = \alpha_1 q_1 \hat{q}_1 = (1-f)\hat{q}_1 \sim \mathcal{O}(1)$. Then, $A$ is determined by the limiting equation

$$\frac{1}{A} = \frac{\sum_{\ell=1}^{L} \frac{A}{\alpha_\ell + A}}{1 - \sum_{\ell=1}^{L} \frac{A}{\alpha_\ell + A}}, \tag{263}$$

and the minimum eigenvalue is given by

$$\mathbb{E}\lambda_{\min} = \frac{1}{1 - \sum_{\ell=1}^{L} \frac{A}{\alpha_\ell + A}} \prod_{\ell=1}^{L} \left(1 + \frac{A}{\alpha_\ell}\right). \tag{264}$$

We can re-write the equation for $A$ as

$$A = \frac{1}{\sum_{\ell=1}^{L} \frac{A}{\alpha_\ell + A}} - 1, \tag{265}$$

and the equation for the minimum eigenvalue as

$$\mathbb{E}\lambda_{\min} = \left(1 + \frac{1}{A}\right)\prod_{\ell=1}^{L}\left(1 + \frac{A}{\alpha_\ell}\right).$$  (266)

For self-consistency with the fact that we should have $q_1 > 0$, we expect to have $A < 0$. Then, letting $B = -A$, we obtain the result claimed in §3.2. Similarly, considering the maximum eigenvalue, we must take $\beta \to -\infty$ through negative values of $\beta$, hence we expect $A \sim \mathcal{O}(1)$ to be positive. Then, we can read off the result reported in §3.2. It is easy to confirm that this condition for the edges of the spectrum is identical to the condition given in equations (70) and (71) of Akemann, Ipsen, and Kieburg [5] for the complex Wishart case, with their $\hat{v}_\ell = \alpha_\ell - 1$ and $\hat{u}_0 = -(A+1)$.

## C.3   Simplifying the recurrence for the replica-uniform order parameters

In this appendix, we solve the saddle point equations for the replica-uniform components of the order parameters in our computation of the minimum and maximum eigenvalues. This analysis amounts to solving a recurrence relation, and follows our approach in [31].
We first eliminate the variables $c_\ell$ by solving the equation

$$\hat{c}_\ell = \frac{\hat{c}_{\ell+1} + c_\ell \hat{q}_{\ell+1}^2}{(1 - q_\ell \hat{q}_{\ell+1})^2}$$  (267)

to obtain

$$c_\ell = \left(\frac{1 - q_\ell \hat{q}_{\ell+1}}{\hat{q}_{\ell+1}}\right)^2 \hat{c}_\ell - \frac{1}{\hat{q}_{\ell+1}^2}\hat{c}_{\ell+1} \qquad\qquad (\ell = 1,\dots,L).$$  (268)

Then, for $\ell = 2,\dots,L$, the equation

$$c_\ell = \frac{\alpha_{\ell-1}}{\alpha_\ell}\frac{c_{\ell-1} + q_{\ell-1}^2 \hat{c}_\ell}{(1 - q_{\ell-1}\hat{q}_\ell)^2}$$  (269)

yields a three-term recurrence

$$\frac{\alpha_{\ell-1}}{\alpha_\ell}\hat{c}_{\ell-1} = \left[\frac{\hat{q}_\ell^2}{\hat{q}_{\ell+1}^2}(1 - q_\ell\hat{q}_{\ell+1})^2 + \frac{\alpha_{\ell-1}}{\alpha_\ell}\frac{1 - q_{\ell-1}^2\hat{q}_\ell^2}{(1 - q_{\ell-1}\hat{q}_\ell)^2}\right]\hat{c}_\ell - \frac{\hat{q}_\ell^2}{\hat{q}_{\ell+1}^2}\hat{c}_{\ell+1}$$  (270)

for $\ell = 2,\dots,L$, with initial difference condition

$$c_1 = \left(\frac{1 - q_1\hat{q}_2}{\hat{q}_2}\right)^2 \hat{c}_1 - \frac{1}{\hat{q}_2^2}\hat{c}_2$$  (271)

and endpoint condition $\hat{c}_{L+1} = 0$. Substituting in the formula

$$q_\ell = \frac{A}{\alpha_\ell \hat{q}_\ell}$$  (272)

and using the recurrence

$$\hat{q}_\ell = \left(1 + \frac{A}{\alpha_\ell}\right)\hat{q}_{\ell+1},$$  (273)

we have

$$q_\ell \hat{q}_{\ell+1} = \frac{A}{\alpha_\ell} \frac{\hat{q}_{\ell+1}}{\hat{q}_\ell} = \frac{A}{\alpha_\ell + A}, \tag{274}$$

hence we obtain the simplified recurrence

$$\frac{\alpha_{\ell-1}}{\alpha_\ell} \hat{c}_{\ell-1} = \frac{\alpha_\ell + \alpha_{\ell-1} + 2A}{\alpha_\ell} \hat{c}_\ell - \left(\frac{\alpha_\ell + A}{\alpha_\ell}\right)^2 \hat{c}_{\ell+1} \tag{275}$$

and the initial difference condition

$$\hat{q}_1^2 c_1 = \hat{c}_1 - \left(\frac{\alpha_1 + A}{\alpha_1}\right)^2 \hat{c}_2. \tag{276}$$

We now further simplify our task by defining new variables $\hat{u}_\ell$ such that

$$\hat{c}_\ell = \alpha_1 \hat{q}_1^2 c_1 \hat{u}_\ell \tag{277}$$

which obey the recurrence

$$\frac{\alpha_{\ell-1}}{\alpha_\ell} \hat{u}_{\ell-1} = \frac{\alpha_\ell + \alpha_{\ell-1} + 2A}{\alpha_\ell} \hat{u}_\ell - \left(\frac{\alpha_\ell + A}{\alpha_\ell}\right)^2 \hat{u}_{\ell+1} \tag{278}$$

for $\ell = 2, \ldots, L$, with the initial difference condition

$$\frac{1}{\alpha_1} = \hat{u}_1 - \left(\frac{\alpha_1 + A}{\alpha_1}\right)^2 \hat{u}_2 \tag{279}$$

and endpoint condition $\hat{u}_{L+1} = 0$. If $L = 1$, we simply have $\hat{u}_1 = 1/\alpha_1$.

To solve this recurrence for $L > 1$, we observe that it can be re-written as

$$\frac{\alpha_\ell + A}{\alpha_\ell} \hat{u}_{\ell+1} - \hat{u}_\ell = \frac{\alpha_{\ell-1}}{\alpha_\ell} \frac{\alpha_\ell}{\alpha_\ell + A} \left[\frac{\alpha_{\ell-1} + A}{\alpha_{\ell-1}} \hat{u}_\ell - \hat{u}_{\ell-1}\right] \tag{280}$$

for $\ell = 2, \ldots, L$. Then, it is easy to see that

$$\frac{\alpha_\ell + A}{\alpha_\ell} \hat{u}_{\ell+1} - \hat{u}_\ell = \frac{\alpha_{\ell-1}}{\alpha_\ell} \frac{\alpha_\ell}{\alpha_\ell + A} \left[\frac{\alpha_{\ell-1} + A}{\alpha_{\ell-1}} \hat{u}_\ell - \hat{u}_{\ell-1}\right] \tag{281}$$

$$= \frac{\alpha_{\ell-1}}{\alpha_\ell} \frac{\alpha_{\ell-2}}{\alpha_{\ell-1}} \frac{\alpha_\ell}{\alpha_\ell + A} \frac{\alpha_{\ell-1}}{\alpha_{\ell-1} + A} \left[\frac{\alpha_{\ell-2} + A}{\alpha_{\ell-2}} \hat{u}_{\ell-1} - \hat{u}_{\ell-2}\right] \tag{282}$$

$$= \frac{\alpha_1}{\alpha_\ell} \frac{\alpha_\ell}{\alpha_\ell + A} \frac{\alpha_{\ell-1}}{\alpha_{\ell-1} + A} \cdots \frac{\alpha_2}{\alpha_2 + A} \left[\frac{\alpha_1 + A}{\alpha_1} \hat{u}_2 - \hat{u}_1\right], \tag{283}$$

hence

$$\hat{u}_\ell = \frac{\alpha_\ell + A}{\alpha_\ell} \hat{u}_{\ell+1} + \frac{1}{\alpha_\ell} \frac{\alpha_\ell}{\alpha_\ell + A} \frac{\alpha_{\ell-1}}{\alpha_{\ell-1} + A} \cdots \frac{\alpha_2}{\alpha_2 + A} [\alpha_1 \hat{u}_1 - (\alpha_1 + A)\hat{u}_2]. \tag{284}$$

By the endpoint condition $\hat{u}_{L+1} = 0$, we then have

$$\hat{u}_L = \frac{1}{\alpha_L} \frac{\alpha_L}{\alpha_L + A} \frac{\alpha_{L-1}}{\alpha_{L-1} + A} \cdots \frac{\alpha_2}{\alpha_2 + A} [\alpha_1 \hat{u}_1 - (\alpha_1 + A)\hat{u}_2], \tag{285}$$

hence

$$\hat{u}_{L-1} = \left( \frac{1}{\alpha_L + A} + \frac{1}{\alpha_{L-1} + A} \right) \frac{\alpha_{L-2}}{\alpha_{L-2} + A} \cdots \frac{\alpha_2}{\alpha_2 + A} [\alpha_1 \hat{u}_1 - (\alpha_1 + A)\hat{u}_2]. \tag{286}$$

Iterating backward, we obtain

$$\hat{u}_\ell = [\alpha_1 \hat{u}_1 - (\alpha_1 + A)\hat{u}_2] \left( \sum_{j=\ell}^{L} \frac{1}{\alpha_j + A} \right) \left( \prod_{j=2}^{\ell-1} \frac{\alpha_j}{\alpha_j + A} \right) \tag{287}$$

for $\ell = 2, \ldots, L$. In particular, we have

$$\hat{u}_2 = [\alpha_1 \hat{u}_1 - (\alpha_1 + A)\hat{u}_2] \sum_{j=2}^{L} \frac{1}{\alpha_j + A}. \tag{288}$$

We now use the initial difference condition to write $\hat{u}_2$ in terms of $\hat{u}_1$,

$$\hat{u}_2 = \left( \frac{\alpha_1}{\alpha_1 + A} \right)^2 \left( \hat{u}_1 - \frac{1}{\alpha_1} \right), \tag{289}$$

which gives a closed equation for $\hat{u}_1$ :

$$\alpha_1 \hat{u}_1 - 1 = (1 + A\hat{u}_1)(\alpha_1 + A) \sum_{j=2}^{L} \frac{1}{\alpha_j + A}, \tag{290}$$

and, for $\ell = 2, \ldots, L$, an expression for $\hat{u}_\ell$ in terms of $\hat{u}_1$:

$$\hat{u}_\ell = (1 + A\hat{u}_1) \left( \sum_{j=\ell}^{L} \frac{1}{\alpha_j + A} \right) \left( \prod_{j=1}^{\ell-1} \frac{\alpha_j}{\alpha_j + A} \right). \tag{291}$$

The equation for $\hat{u}_1$ simplifies to

$$\frac{\hat{u}_1}{1 + A\hat{u}_1} = \sum_{j=1}^{L} \frac{1}{\alpha_j + A}, \tag{292}$$

which yields

$$\hat{u}_1 = \frac{\sum_{j=1}^{L} \frac{1}{\alpha_j + A}}{1 - A \sum_{j=1}^{L} \frac{1}{\alpha_j + A}}. \tag{293}$$

If $L = 1$, this recovers the expected result that $\hat{u}_1 = 1/\alpha_1$. From this result, we have

$$\hat{c}_1 = \alpha_1 \hat{q}_1^2 c_1 \hat{u}_1 \tag{294}$$

$$= \frac{\hat{q}_1}{q_1} c_1 \frac{\sum_{j=1}^{L} \frac{A}{\alpha_j + A}}{1 - \sum_{j=1}^{L} \frac{A}{\alpha_j + A}} \tag{295}$$

which will allow us to obtain a self-consistent equation given the definition of $\hat{c}_1$ in terms of $f$.

Recalling from §C.2 that $\mathbb{E}\lambda_{\min}$ is given in terms of $c_L$, we use the condition

$$\hat{q}_L^2 c_L = \hat{c}_L - \left(\frac{\alpha_L + A}{\alpha_L}\right)^2 \hat{c}_{L+1} \tag{296}$$

to obtain

$$c_L = \frac{1}{\hat{q}_L^2}\hat{c}_L = \frac{\alpha_1 c_1}{\alpha_L}\frac{\hat{q}_1^2}{\hat{q}_L^2}(1 + A\hat{u}_1)\left(\prod_{j=1}^{L}\frac{\alpha_j}{\alpha_j + A}\right) \tag{297}$$

as $\hat{c}_{L+1} = 0$ and $\hat{c}_\ell = \alpha_1\hat{q}_1^2 c_1\hat{u}_\ell$ by definition, and

$$\hat{u}_L = \frac{1}{\alpha_L}(1 + A\hat{u}_1)\left(\prod_{j=1}^{L}\frac{\alpha_j}{\alpha_j + A}\right). \tag{298}$$

Substituting in the value of $\hat{u}_1$, we find that

$$c_L = \frac{\alpha_1 c_1}{\alpha_L}\frac{\hat{q}_1^2}{\hat{q}_L^2}\left(\prod_{j=1}^{L}\frac{\alpha_j}{\alpha_j + A}\right)\frac{1}{1 - \sum_{j=1}^{L}\frac{A}{a_j + A}}. \tag{299}$$

These are the results reported in §C.2.

## D   Computing the extremal eigenvalues for row-structured factors

### D.1   Step I: Evaluating the moments of the partition function

As in our study of the unstructured case, we consider the moments of the partition function of the spherical spin glass:

$$\mathbb{E}Z^m = \int \prod_a d\mathbf{w}^a \left[\prod_{a=1}^{m}\delta\left(1 - \frac{1}{n_0}\|\mathbf{w}^a\|^2\right)\right]\mathbb{E}\exp\left(\frac{-\beta}{2n_L\cdots n_1}\sum_{a=1}^{m}(\mathbf{w}^a)^\top\mathbf{X}_1^\top\cdots\mathbf{X}_L^\top\mathbf{X}_L\cdots\mathbf{X}_1\mathbf{w}^a\right). \tag{300}$$

Again, we can integrate out the matrices $\mathbf{X}_\ell$ iteratively, introducing the order parameters

$$C_\ell^{ab} \equiv \frac{1}{n_1\cdots n_\ell}(\mathbf{w}^a)^\top\mathbf{X}_1^\top\cdots\mathbf{X}_{\ell-1}^\top\mathbf{X}_{\ell-1}\cdots\mathbf{X}_1\mathbf{w}^b, \tag{301}$$

and the modified boundary condition $\hat{\mathbf{C}}_{L+1} = -\beta\mathbf{I}_m$. Then, iterating backwards until only the vectors $\mathbf{w}^a$ remain, we have

$$\mathbb{E}Z^m = \int \frac{d\mathbf{C}_2\,d\hat{\mathbf{C}}_2}{(4\pi i/n_1)^{m(m+1)/2}}\cdots\int\frac{d\mathbf{C}_L\,d\hat{\mathbf{C}}_L}{(4\pi i/n_L)^{m(m+1)/2}}$$
$$\exp\left(-\frac{1}{2}\sum_{\ell=2}^{L}n_\ell\left[\text{tr}(\mathbf{C}_\ell\hat{\mathbf{C}}_\ell) + \frac{1}{n_\ell}\log\det[\mathbf{I}_{mn_\ell} - (\mathbf{C}_\ell\hat{\mathbf{C}}_{\ell+1})\otimes\Sigma_\ell]\right]\right)$$
$$\times \int\prod_{a=1}^{m}d\mathbf{w}^a\left[\prod_{a=1}^{m}\delta\left(1 - \frac{1}{n_0}\|\mathbf{w}^a\|^2\right)\right]\det(\mathbf{I}_{mn_1} - (\mathbf{C}_1\hat{\mathbf{C}}_2)\otimes\Sigma_1)^{-n_1/2}, \tag{302}$$

where we recall that

$$C_1^{ab} = \frac{1}{n_1}(\mathbf{w}^a)^\top \mathbf{w}^b. \tag{303}$$

As in the unstructured case, the spherical constraint means that it is useful to introduce order parameters

$$F^{ab} = \frac{1}{n_0}(\mathbf{w}^a)^\top \mathbf{w}^b \tag{304}$$

via Fourier representations of the $\delta$-distribution, such that $F^{aa} = 1$ and $\mathbf{C}_1 = \mathbf{F}/\alpha_1$. It is this step—and, concretely, the spherical constraint—that would be difficult to tackle in the presence of column-wise correlations in the first factor, as one would have $C_1^{ab} = (\mathbf{w}^a)^\top \mathbf{\Gamma}_1 \mathbf{w}^b/n_1$, which is not immediately compatible with the spherical constraint.

Then, after evaluating the remaining unconstrained Gaussian integral over $\mathbf{w}^a$, we obtain

$$\mathbb{E}Z^m \propto \int \frac{d\mathbf{F}\,d\hat{\mathbf{F}}}{(4\pi i/n_0)^{m(m+1)/2}} \int \frac{d\mathbf{C}_2\,d\hat{\mathbf{C}}_2}{(4\pi i/n_2)^{m(m+1)/2}} \cdots \int \frac{d\mathbf{C}_L\,d\hat{\mathbf{C}}_L}{(4\pi i/n_L)^{m(m+1)/2}} \exp\left(\frac{n_0 m}{2}S\right) \tag{305}$$

for

$$S(\mathbf{F}, \hat{\mathbf{F}}, \mathbf{C}_2, \hat{\mathbf{C}}_2, \cdots, \mathbf{C}_L, \hat{\mathbf{C}}_L) = \frac{1}{m}\operatorname{tr}(\mathbf{F}\hat{\mathbf{F}}) - \frac{1}{m}\log\det(\hat{\mathbf{F}}) - \frac{1}{m}\alpha_1\frac{1}{n_1}\log\det(\mathbf{I}_{mn_1} - \alpha_1^{-1}(\mathbf{F}\hat{\mathbf{C}}_2) \otimes \mathbf{\Sigma}_1)$$

$$- \frac{1}{m}\sum_{\ell=2}^{L}\alpha_\ell\left[\operatorname{tr}(\mathbf{C}_\ell\hat{\mathbf{C}}_\ell) + \frac{1}{n_\ell}\log\det[\mathbf{I}_{mn_\ell} - (\mathbf{C}_\ell\hat{\mathbf{C}}_{\ell+1}) \otimes \mathbf{\Sigma}_\ell]\right]. \tag{306}$$

Again, under the assumption that the spectra of the matrices $\mathbf{\Sigma}_\ell$ are sufficiently generic, the action $S$ is $\mathcal{O}(1)$, and the integral can be evaluated using the method of steepest descent.

## D.2 Step II: The replica-symmetric saddle point equations

As elsewhere, we make an RS *Ansatz*

$$\mathbf{F} = (1-f)\mathbf{I}_m + f\mathbf{1}_m\mathbf{1}_m^\top \tag{307}$$

$$\hat{\mathbf{F}} = (\hat{F} - \hat{f})\mathbf{I}_m + \hat{f}\mathbf{1}_m\mathbf{1}_m^\top \tag{308}$$

$$\mathbf{C}_\ell = q_\ell\mathbf{I}_m + c_\ell\mathbf{1}_m\mathbf{1}_m^\top \qquad\qquad (\ell = 2,\ldots,L) \tag{309}$$

$$\hat{\mathbf{C}}_\ell = \hat{q}_\ell\mathbf{I}_m + \hat{c}_\ell\mathbf{1}_m\mathbf{1}_m^\top \qquad\qquad (\ell = 2,\ldots,L). \tag{310}$$

Combining our analysis of the extremal eigenvalues in the unstructured case with our analysis of the Stieltjes transform in the structured case, we have

$$\lim_{m\to 0} S = \hat{F} - f\hat{f} - \log(\hat{F} - \hat{f}) - \frac{\hat{f}}{\hat{F} - \hat{f}}$$

$$- \alpha_1\left(\mathbb{E}_{\sigma_1}\log(1 - q_1\hat{q}_2\sigma_1) - (q_1\hat{c}_2 + c_1\hat{q}_2)\mathbb{E}_{\sigma_1}\left[\frac{\sigma_1}{1 - q_1\hat{q}_2\sigma_1}\right]\right)$$

$$- \sum_{\ell=2}^{L}\alpha_\ell\left(q_\ell\hat{q}_\ell + q_\ell\hat{c}_\ell + c_\ell\hat{q}_\ell + \mathbb{E}_{\sigma_\ell}\log(1 - q_\ell\hat{q}_{\ell+1}\sigma_\ell)\right.$$

$$\left. - (q_\ell\hat{c}_{\ell+1} + c_\ell\hat{q}_{\ell+1})\mathbb{E}_{\sigma_\ell}\left[\frac{\sigma_\ell}{1 - q_\ell\hat{q}_{\ell+1}\sigma_\ell}\right]\right), \tag{311}$$

where we recall the endpoint condition $\hat{q}_{L+1} = -\beta$, $\hat{c}_{L+1} = 0$ and, for brevity, we define $q_1 = \alpha_1^{-1}(1-f)$ and $c_1 = \alpha_1^{-1}f$.

Then, by comparison with our previous results, we can read off that, after eliminating $\hat{F}$ and $\hat{f}$, the saddle point equations can be written as

$$\hat{q}_\ell = \hat{q}_{\ell+1}\mathbb{E}_{\sigma_\ell}\left[\frac{\sigma_\ell}{1-q_\ell\hat{q}_{\ell+1}\sigma_\ell}\right] \qquad (\ell = 2,\ldots,L) \qquad (312)$$

$$q_\ell = \frac{\alpha_{\ell-1}}{\alpha_\ell}q_{\ell-1}\mathbb{E}_{\sigma_{\ell-1}}\left[\frac{\sigma_{\ell-1}}{1-q_{\ell-1}\hat{q}_\ell\sigma_{\ell-1}}\right] \qquad (\ell = 2,\ldots,L) \qquad (313)$$

$$\hat{c}_\ell = \hat{c}_{\ell+1}\mathbb{E}_{\sigma_\ell}\left[\frac{\sigma_\ell}{1-q_\ell\hat{q}_{\ell+1}\sigma_\ell}\right]$$
$$+ (q_\ell\hat{c}_{\ell+1} + c_\ell\hat{q}_{\ell+1})\hat{q}_{\ell+1}\mathbb{E}_{\sigma_\ell}\left[\left(\frac{\sigma_\ell}{1-q_\ell\hat{q}_{\ell+1}\sigma_\ell}\right)^2\right] \qquad (\ell = 1,\ldots,L) \qquad (314)$$

$$c_\ell = \frac{\alpha_{\ell-1}}{\alpha_\ell}c_{\ell-1}\mathbb{E}_{\sigma_{\ell-1}}\left[\frac{\sigma_{\ell-1}}{1-q_{\ell-1}\hat{q}_\ell\sigma_{\ell-1}}\right]$$
$$+ \frac{\alpha_{\ell-1}}{\alpha_\ell}(q_{\ell-1}\hat{c}_\ell + c_{\ell-1}\hat{q}_\ell)q_{\ell-1}\mathbb{E}_{\sigma_{\ell-1}}\left[\left(\frac{\sigma_{\ell-1}}{1-q_{\ell-1}\hat{q}_\ell\sigma_{\ell-1}}\right)^2\right] \qquad (\ell = 2,\ldots,L), \qquad (315)$$

where we have the definitions

$$q_1 \equiv \alpha_1^{-1}(1-f) \qquad (316)$$
$$c_1 \equiv \alpha_1^{-1}f \qquad (317)$$
$$\hat{c}_1 \equiv \frac{f}{(1-f)^2} \qquad (318)$$

and the endpoint conditions

$$\hat{q}_{L+1} = -\beta \qquad (319)$$
$$\hat{c}_{L+1} = 0. \qquad (320)$$

Moreover, we have

$$\mathbb{E}\lambda_{\min} = -\lim_{\beta\to\infty}\lim_{m\to 0}\frac{\partial S}{\partial \beta} \qquad (321)$$

$$= \alpha_L \lim_{\beta\to\infty}\frac{\partial}{\partial\beta}\left(\mathbb{E}_{\sigma_L}\log(1+\beta q_L\sigma_L) + \beta c_L\mathbb{E}_{\sigma_L}\left[\frac{\sigma_L}{1+\beta q_L\sigma_L}\right]\right) \qquad (322)$$

$$= \alpha_L \lim_{\beta\to\infty}\left(\mathbb{E}_{\sigma_L}\left[\frac{q_L\sigma_L}{1+\beta q_L\sigma_L}\right] + c_L\mathbb{E}_{\sigma_L}\left[\frac{\sigma_L}{1+\beta q_L\sigma_L}\right] - c_L\beta q_L\mathbb{E}_{\sigma_L}\left[\left(\frac{\sigma_L}{1+\beta q_L\sigma_L}\right)^2\right]\right) \qquad (323)$$

where the order parameters are to be evaluated at their saddle point values. Our task is therefore to solve the saddle point equations in the zero temperature limit.

To solve for the replica-uniform components, we define an auxiliary variable $\hat{q}_1$ by

$$\hat{q}_1 = \hat{q}_2\mathbb{E}_{\sigma_1}\left[\frac{\sigma_1}{1-q_1\hat{q}_2\sigma_1}\right], \qquad (324)$$

such that we have the same system of equations as in our analysis of the Stieltjes transform. Writing

$$A = \alpha_1 q_1 \hat{q}_1, \tag{325}$$

we have

$$q_\ell \hat{q}_\ell = \frac{A}{\alpha_\ell} \tag{326}$$

for all $\ell = 1, \ldots, L$, and the expression

$$\frac{1}{q_\ell \hat{q}_{\ell+1}} = M_{\Sigma_\ell}^{-1}\left(\frac{A}{\alpha_\ell}\right) \tag{327}$$

for all $\ell = 1, \ldots, L$ in terms of the moment generating functions of the correlation matrices.

Then, using the boundary condition $\hat{q}_{L+1} = -\beta$, we have

$$\frac{1}{q_L} = -\beta M_{\Sigma_L}^{-1}\left(\frac{A}{\alpha_L}\right) \tag{328}$$

For $\ell = 1, \ldots, L$, we multiply through by $q_{\ell+1}\hat{q}_{\ell+1}$ to obtain

$$\frac{q_{\ell+1}}{q_\ell} = \frac{A}{\alpha_{\ell+1}} M_{\Sigma_\ell}^{-1}\left(\frac{A}{\alpha_\ell}\right), \tag{329}$$

hence we can iterate backward to obtain

$$\frac{q_L}{q_\ell} = \frac{q_{\ell+1}}{q_\ell}\frac{q_{\ell+2}}{q_{\ell+1}}\cdots\frac{q_L}{q_{L-1}} \tag{330}$$

$$= \prod_{j=\ell}^{L-1} \frac{A}{\alpha_{j+1}} M_{\Sigma_j}^{-1}\left(\frac{A}{\alpha_j}\right) \tag{331}$$

whence

$$\frac{1}{q_\ell} = -\beta \frac{\alpha_\ell}{A} \prod_{j=\ell}^{L} \frac{A}{\alpha_j} M_{\Sigma_j}^{-1}\left(\frac{A}{\alpha_j}\right) \tag{332}$$

and

$$\hat{q}_\ell = -\beta \prod_{j=\ell}^{L} \frac{A}{\alpha_j} M_{\Sigma_j}^{-1}\left(\frac{A}{\alpha_j}\right). \tag{333}$$

The equations for replica-uniform components can be simplified after a bit of algebra, which we defer to §D.3. This computation results in the condition

$$\hat{c}_1 = c_1 \frac{\hat{q}_1}{q_1} \frac{\sum_{j=1}^{L}(\mu_j - 1)/\mu_j}{1 - \sum_{k=1}^{L}(\mu_k - 1)/\mu_k} \tag{334}$$

and the equation

$$c_L = \frac{q_L}{q_1} c_1 \frac{1}{\mu_L} \frac{1}{1 - \sum_{k=1}^{L}(\mu_k - 1)/\mu_k}, \tag{335}$$

where we define

$$\mu_\ell = -\frac{\alpha_\ell}{A}\frac{M_{\Sigma_\ell}^{-1}(A/\alpha_\ell)}{(M_{\Sigma_\ell}^{-1})'(A/\alpha_\ell)} \tag{336}$$

to express

$$\mathbb{E}_{\sigma_\ell}\left[\left(\frac{q_\ell\hat{q}_{\ell+1}\sigma_\ell}{1-q_\ell\hat{q}_{\ell+1}\sigma_\ell}\right)^2\right] = \frac{A}{\alpha_\ell}(\mu_\ell-1) \tag{337}$$

in terms of $M_{\Sigma_\ell}$.

When combined with the definitions

$$q_1 \equiv \alpha_1^{-1}(1-f) \tag{338}$$

$$c_1 \equiv \alpha_1^{-1}f \tag{339}$$

$$\hat{c}_1 \equiv \frac{f}{(1-f)^2}, \tag{340}$$

the equation for $\hat{c}_1$ gives a closed equation for $A = \alpha_1 q_1 \hat{q}_1 = (1-f)\hat{q}_1$:

$$\frac{1}{A} = \frac{\sum_{\ell=1}^{L}(\mu_\ell-1)/\mu_\ell}{1-\sum_{\ell=1}^{L}(\mu_\ell-1)/\mu_\ell}. \tag{341}$$

Using the results of §D.3, we may re-write the expression obtained above for $\mathbb{E}\lambda_{\min}$ in terms of $M_{\Sigma_L}$ and $\mu_L$:

$$\mathbb{E}\lambda_{\min} = \alpha_L \lim_{\beta\to\infty}\left(\mathbb{E}_{\sigma_L}\left[\frac{q_L\sigma_L}{1+\beta q_L\sigma_L}\right] + c_L\mathbb{E}_{\sigma_L}\left[\frac{\sigma_L}{1+\beta q_L\sigma_L}\right] - c_L\beta q_L\mathbb{E}_{\sigma_L}\left[\left(\frac{\sigma_L}{1+\beta q_L\sigma_L}\right)^2\right]\right) \tag{342}$$

$$= \alpha_L \lim_{\beta\to\infty}\left(-\frac{1}{\beta}\frac{A}{\alpha_L} - \frac{1}{\beta q_L}\frac{A}{\alpha_L}c_L - c_L\frac{1}{\beta q_L}\frac{A}{\alpha_L}(\mu_L-1)\right) \tag{343}$$

$$= -\lim_{\beta\to\infty}\left(\frac{A\mu_L}{\beta q_L}c_L + \frac{A}{\beta}\right), \tag{344}$$

as

$$\mathbb{E}_{\sigma_L}\left[\frac{q_L\sigma_L}{1+\beta q_L\sigma_L}\right] = -\frac{1}{\beta}M_{\Sigma_L}\left(\frac{1}{q_L\hat{q}_{L+1}}\right) \tag{345}$$

$$= -\frac{1}{\beta}\frac{A}{\alpha_L} \tag{346}$$

and

$$\mathbb{E}_{\sigma_L}\left[\left(\frac{\beta q_L}{1+\beta q_L}\right)^2\right] = \mathbb{E}_{\sigma_L}\left[\left(\frac{q_L\hat{q}_{L+1}\sigma_\ell}{1-q_L\hat{q}_{L+1}\sigma_L}\right)^2\right] \tag{347}$$

$$= \frac{A}{\alpha_L}(\mu_L-1). \tag{348}$$

Given the results we have obtained thus far, we expect that $q_\ell \sim \mathcal{O}(1/\beta)$, $\hat{q}_\ell \sim \mathcal{O}(\beta)$, $A \sim \mathcal{O}(1)$, and

$$c_1 = \frac{f}{\alpha_1} = \frac{1}{\alpha_1} - q_1 = \frac{1}{\alpha_1} + \mathcal{O}(1/\beta) \tag{349}$$

as $\beta \to \infty$. Then,

$$\mathbb{E}\lambda_{\min} = -\lim_{\beta \to \infty} \frac{A\mu_L}{\beta q_L} c_L \tag{350}$$

$$= -\lim_{\beta \to \infty} \frac{A\mu_L}{\beta q_1} c_1 \frac{1}{\mu_L} \frac{1}{1 - \sum_{k=1}^{L}(\mu_k - 1)/\mu_k} \tag{351}$$

$$= \frac{1}{1 - \sum_{k=1}^{L}(\mu_k - 1)/\mu_k} \prod_{j=1}^{L} \frac{A}{\alpha_j} M_{\Sigma_j}^{-1}\left(\frac{A}{\alpha_j}\right), \tag{352}$$

where we use the fact that

$$\frac{1}{q_1} = -\beta \frac{\alpha_1}{A} \prod_{j=1}^{L} \frac{A}{\alpha_j} M_{\Sigma_j}^{-1}\left(\frac{A}{\alpha_j}\right). \tag{353}$$

We thus have found that

$$\mathbb{E}\lambda_{\min} = \left(1 + \frac{1}{A}\right) \prod_{\ell=1}^{L} \frac{A}{\alpha_\ell} M_{\Sigma_\ell}^{-1}\left(\frac{A}{\alpha_\ell}\right), \tag{354}$$

where $A$ satisfies

$$\frac{1}{A} = \frac{\sum_{\ell=1}^{L}(\mu_\ell(A) - 1)/\mu_\ell(A)}{1 - \sum_{\ell=1}^{L}(\mu_\ell(A) - 1)/\mu_\ell(A)}, \tag{355}$$

or, equivalently,

$$A = \frac{1}{\sum_{\ell=1}^{L}(\mu_\ell(A) - 1)/\mu_\ell(A)} - 1, \tag{356}$$

for

$$\mu_\ell(A) = -\frac{\alpha_\ell}{A} \frac{M_{\Sigma_\ell}^{-1}(A/\alpha_\ell)}{(M_{\Sigma_\ell}^{-1})'(A/\alpha_\ell)}. \tag{357}$$

This is the result claimed in §3.3.

## D.3   Simplifying the recurrence for the replica-uniform order parameters

In this section, we simplify the recurrence, derived in §D.2 that determines the replica-uniform order parameters in the extremal eigenvalue computation for row-structured matrices. We want

to simplify the linear system

$$
\hat{c}_\ell = \hat{c}_{\ell+1}\mathbb{E}_{\sigma_\ell}\left[\frac{\sigma_\ell}{1-q_\ell\hat{q}_{\ell+1}\sigma_\ell}\right]
$$
$$
+ (q_\ell\hat{c}_{\ell+1}+c_\ell\hat{q}_{\ell+1})\hat{q}_{\ell+1}\mathbb{E}_{\sigma_\ell}\left[\left(\frac{\sigma_\ell}{1-q_\ell\hat{q}_{\ell+1}\sigma_\ell}\right)^2\right] \qquad (\ell=1,\dots,L) \tag{358}
$$

$$
c_\ell = \frac{\alpha_{\ell-1}}{\alpha_\ell}c_{\ell-1}\mathbb{E}_{\sigma_{\ell-1}}\left[\frac{\sigma_{\ell-1}}{1-q_{\ell-1}\hat{q}_\ell\sigma_{\ell-1}}\right]
$$
$$
+ \frac{\alpha_{\ell-1}}{\alpha_\ell}(q_{\ell-1}\hat{c}_\ell+c_{\ell-1}\hat{q}_\ell)q_{\ell-1}\mathbb{E}_{\sigma_{\ell-1}}\left[\left(\frac{\sigma_{\ell-1}}{1-q_{\ell-1}\hat{q}_\ell\sigma_{\ell-1}}\right)^2\right] \qquad (\ell=2,\dots,L) \tag{359}
$$

for fixed $c_1$, using the boundary condition $\hat{c}_{L+1}=0$.

Regrouping terms, we have

$$
q_\ell\hat{q}_{\ell+1}\hat{c}_\ell = \left(\mathbb{E}_{\sigma_\ell}\left[\frac{q_\ell\hat{q}_{\ell+1}\sigma_\ell}{1-q_\ell\hat{q}_{\ell+1}\sigma_\ell}\right]+\mathbb{E}_{\sigma_\ell}\left[\left(\frac{q_\ell\hat{q}_{\ell+1}\sigma_\ell}{1-q_\ell\hat{q}_{\ell+1}\sigma_\ell}\right)^2\right]\right)\hat{c}_{\ell+1}
$$
$$
+ \left(\frac{\hat{q}_{\ell+1}}{q_\ell}\mathbb{E}_{\sigma_\ell}\left[\left(\frac{q_\ell\hat{q}_{\ell+1}\sigma_\ell}{1-q_\ell\hat{q}_{\ell+1}\sigma_\ell}\right)^2\right]\right)c_\ell \qquad (\ell=1,\dots,L) \tag{360}
$$

$$
\frac{\alpha_\ell}{\alpha_{\ell-1}}q_{\ell-1}\hat{q}_\ell c_\ell = \left(\mathbb{E}_{\sigma_{\ell-1}}\left[\frac{q_{\ell-1}\hat{q}_\ell\sigma_{\ell-1}}{1-q_{\ell-1}\hat{q}_\ell\sigma_{\ell-1}}\right]+\mathbb{E}_{\sigma_{\ell-1}}\left[\left(\frac{q_{\ell-1}\hat{q}_\ell\sigma_{\ell-1}}{1-q_{\ell-1}\hat{q}_\ell\sigma_{\ell-1}}\right)^2\right]\right)c_{\ell-1}
$$
$$
+ \left(\frac{q_{\ell-1}}{\hat{q}_\ell}\mathbb{E}_{\sigma_{\ell-1}}\left[\left(\frac{q_{\ell-1}\hat{q}_\ell\sigma_{\ell-1}}{1-q_{\ell-1}\hat{q}_\ell\sigma_{\ell-1}}\right)^2\right]\right)\hat{c}_\ell \qquad (\ell=2,\dots,L). \tag{361}
$$

As in our analysis of the replica nonuniform components, we write the expectations in terms of the spectral generating function. As noted before, we have

$$
\mathbb{E}_{\sigma_\ell}\left[\frac{q_\ell\hat{q}_{\ell+1}\sigma_\ell}{1-q_\ell\hat{q}_{\ell+1}\sigma_\ell}\right] = M_{\Sigma_\ell}\left(\frac{1}{q_\ell\hat{q}_{\ell+1}}\right) \tag{362}
$$

for all $\ell=1,\dots,L$. Similarly,

$$
\mathbb{E}_{\sigma_\ell}\left[\left(\frac{q_\ell\hat{q}_{\ell+1}\sigma_\ell}{1-q_\ell\hat{q}_{\ell+1}\sigma_\ell}\right)^2\right] = -M_{\Sigma_\ell}\left(\frac{1}{q_\ell\hat{q}_{\ell+1}}\right) - \frac{1}{q_\ell\hat{q}_{\ell+1}}M'_{\Sigma_\ell}\left(\frac{1}{q_\ell\hat{q}_{\ell+1}}\right), \tag{363}
$$

where $M'_{\Sigma_\ell}(z)$ denotes the first derivative of $M_{\Sigma_\ell}$ with respect to its argument. From our analysis above, letting

$$
A = \alpha_1 q_1 \hat{q}_1, \tag{364}
$$

we have

$$
M_{\Sigma_\ell}\left(\frac{1}{q_\ell\hat{q}_{\ell+1}}\right) = \frac{A}{\alpha_\ell} \tag{365}
$$

for all $\ell=1,\dots,L$. Thus,

$$
\mathbb{E}_{\sigma_\ell}\left[\frac{q_\ell\hat{q}_{\ell+1}\sigma_\ell}{1-q_\ell\hat{q}_{\ell+1}\sigma_\ell}\right] = \frac{A}{\alpha_\ell}, \tag{366}
$$

while

$$\mathbb{E}_{\sigma_\ell}\left[\left(\frac{q_\ell\hat{q}_{\ell+1}\sigma_\ell}{1-q_\ell\hat{q}_{\ell+1}\sigma_\ell}\right)^2\right] = -\frac{A}{\alpha_\ell} - \frac{1}{q_\ell\hat{q}_{\ell+1}}M'_{\Sigma_\ell}\left(M_{\Sigma_\ell}^{-1}\left(\frac{A}{\alpha_\ell}\right)\right). \tag{367}$$

Writing

$$\mu_\ell \equiv -\frac{\alpha_\ell}{A}\frac{1}{q_\ell\hat{q}_{\ell+1}}M'_{\Sigma_\ell}\left(M_{\Sigma_\ell}^{-1}\left(\frac{A}{\alpha_\ell}\right)\right) \tag{368}$$

$$= -\frac{\alpha_\ell}{A}M_{\Sigma_\ell}^{-1}\left(\frac{A}{\alpha_\ell}\right)M'_{\Sigma_\ell}\left(M_{\Sigma_\ell}^{-1}\left(\frac{A}{\alpha_\ell}\right)\right) \tag{369}$$

$$= -\frac{\alpha_\ell}{A}\frac{M_{\Sigma_\ell}^{-1}(A/\alpha_\ell)}{(M_{\Sigma_\ell}^{-1})'(A/\alpha_\ell)} \tag{370}$$

such that

$$\mathbb{E}_{\sigma_\ell}\left[\left(\frac{q_\ell\hat{q}_{\ell+1}\sigma_\ell}{1-q_\ell\hat{q}_{\ell+1}\sigma_\ell}\right)^2\right] = \frac{A}{\alpha_\ell}(\mu_\ell - 1), \tag{371}$$

we have

$$q_\ell\hat{q}_{\ell+1}\hat{c}_\ell = \frac{A}{\alpha_\ell}\mu_\ell\hat{c}_{\ell+1} + \frac{\hat{q}_{\ell+1}}{q_\ell}\frac{A}{\alpha_\ell}(\mu_\ell - 1)c_\ell \qquad (\ell = 1,\ldots,L) \tag{372}$$

$$q_{\ell-1}\hat{q}_\ell c_\ell = \frac{A}{\alpha_\ell}\mu_{\ell-1}c_{\ell-1} + \frac{q_{\ell-1}}{\hat{q}_\ell}\frac{A}{\alpha_\ell}(\mu_{\ell-1} - 1)\hat{c}_\ell \qquad (\ell = 2,\ldots,L). \tag{373}$$

Using the fact that

$$q_\ell\hat{q}_\ell = \frac{A}{\alpha_\ell}, \tag{374}$$

we can re-write this as

$$\frac{\hat{q}_{\ell+1}}{\hat{q}_\ell}\hat{c}_\ell = \mu_\ell\hat{c}_{\ell+1} + \frac{\hat{q}_{\ell+1}}{q_\ell}(\mu_\ell - 1)c_\ell \qquad (\ell = 1,\ldots,L) \tag{375}$$

$$\frac{q_{\ell-1}}{q_\ell}c_\ell = \mu_{\ell-1}c_{\ell-1} + \frac{q_{\ell-1}}{\hat{q}_\ell}(\mu_{\ell-1} - 1)\hat{c}_\ell \qquad (\ell = 2,\ldots,L). \tag{376}$$

We now solve the first equation for $c_\ell$, yielding

$$c_\ell = \frac{1}{\mu_\ell - 1}\frac{q_\ell}{\hat{q}_{\ell+1}}\left(\frac{\hat{q}_{\ell+1}}{\hat{q}_\ell}\hat{c}_\ell - \mu_\ell\hat{c}_{\ell+1}\right) \tag{377}$$

for $\ell = 1,\ldots,L$. Substituting this into the second equation, we have

$$\frac{1}{\mu_\ell - 1}\frac{\hat{q}_\ell}{\hat{q}_{\ell+1}}\left(\frac{\hat{q}_{\ell+1}}{\hat{q}_\ell}\hat{c}_\ell - \mu_\ell\hat{c}_{\ell+1}\right) = \frac{\mu_{\ell-1}}{\mu_{\ell-1} - 1}\left(\frac{\hat{q}_\ell}{\hat{q}_{\ell-1}}\hat{c}_{\ell-1} - \mu_{\ell-1}\hat{c}_\ell\right) + (\mu_{\ell-1} - 1)\hat{c}_\ell. \tag{378}$$

Expanding and adding $\hat{c}_\ell$ to both sides, we have

$$\frac{\mu_\ell}{\mu_\ell - 1}\hat{c}_\ell - \frac{\mu_\ell}{\mu_\ell - 1}\frac{\hat{q}_\ell}{\hat{q}_{\ell+1}}\hat{c}_{\ell+1} = \frac{\mu_{\ell-1}}{\mu_{\ell-1} - 1}\frac{\hat{q}_\ell}{\hat{q}_{\ell-1}}\hat{c}_{\ell-1} - \frac{\mu_{\ell-1}}{\mu_{\ell-1} - 1}\hat{c}_\ell, \tag{379}$$

or

$$\frac{\mu_\ell}{\mu_\ell - 1}\left(\frac{\hat{q}_{\ell+1}}{\hat{q}_\ell}\hat{c}_\ell - \hat{c}_{\ell+1}\right) = \frac{\hat{q}_{\ell+1}}{\hat{q}_\ell}\frac{\mu_{\ell-1}}{\mu_{\ell-1}-1}\left(\frac{\hat{q}_\ell}{\hat{q}_{\ell-1}}\hat{c}_{\ell-1} - \hat{c}_\ell\right). \tag{380}$$

This can be iterated backward, yielding

$$\frac{\mu_\ell}{\mu_\ell - 1}\left(\frac{\hat{q}_{\ell+1}}{\hat{q}_\ell}\hat{c}_\ell - \hat{c}_{\ell+1}\right) = \frac{\hat{q}_{\ell+1}}{\hat{q}_\ell}\frac{\mu_{\ell-1}}{\mu_{\ell-1}-1}\left(\frac{\hat{q}_\ell}{\hat{q}_{\ell-1}}\hat{c}_{\ell-1} - \hat{c}_\ell\right) \tag{381}$$

$$= \frac{\hat{q}_{\ell+1}}{\hat{q}_{\ell-1}}\frac{\mu_{\ell-2}}{\mu_{\ell-2}-1}\left(\frac{\hat{q}_{\ell-1}}{\hat{q}_{\ell-2}}\hat{c}_{\ell-2} - \hat{c}_{\ell-1}\right) \tag{382}$$

$$\vdots \tag{383}$$

$$= \frac{\hat{q}_{\ell+1}}{\hat{q}_2}\frac{\mu_1}{\mu_1-1}\left(\frac{\hat{q}_2}{\hat{q}_1}\hat{c}_1 - \hat{c}_2\right) \tag{384}$$

for all $\ell = 2, \ldots, L$. Re-arranging, this gives

$$\hat{c}_\ell = \frac{\hat{q}_\ell}{\hat{q}_{\ell+1}}\hat{c}_{\ell+1} + \frac{\hat{q}_\ell}{\hat{q}_2}\frac{\mu_\ell - 1}{\mu_\ell}\frac{\mu_1}{\mu_1-1}\left(\frac{\hat{q}_2}{\hat{q}_1}\hat{c}_1 - \hat{c}_2\right) \tag{385}$$

for $\ell = 2, \ldots, L$. Using the boundary condition $\hat{c}_{L+1} = 0$, we then have

$$\hat{c}_L = \frac{\hat{q}_L}{\hat{q}_2}\frac{\mu_L - 1}{\mu_L}\frac{\mu_1}{\mu_1-1}\left(\frac{\hat{q}_2}{\hat{q}_1}\hat{c}_1 - \hat{c}_2\right), \tag{386}$$

hence we obtain

$$\hat{c}_\ell = \frac{\hat{q}_\ell}{\hat{q}_2}\frac{\mu_1}{\mu_1-1}\left(\frac{\hat{q}_2}{\hat{q}_1}\hat{c}_1 - \hat{c}_2\right)\sum_{j=\ell}^{L}\frac{\mu_j - 1}{\mu_j} \tag{387}$$

for $\ell = 2, \ldots, L$. To obtain a closed equation for $\hat{c}_1$, we use the condition

$$c_1 = \frac{1}{\mu_1-1}\frac{q_1}{\hat{q}_2}\left(\frac{\hat{q}_2}{\hat{q}_1}\hat{c}_1 - \mu_1\hat{c}_2\right). \tag{388}$$

As in our previous analysis, it is useful to make the change of variables

$$\hat{c}_\ell = \alpha_1\hat{q}_1^2 c_1\hat{u}_\ell, \tag{389}$$

which satisfy

$$\hat{u}_\ell = \frac{\hat{q}_\ell}{\hat{q}_2}\frac{\mu_1}{\mu_1-1}\left(\frac{\hat{q}_2}{\hat{q}_1}\hat{u}_1 - \hat{u}_2\right)\sum_{j=\ell}^{L}\frac{\mu_j - 1}{\mu_j} \tag{390}$$

for $\ell = 2, \ldots, L$, and the condition

$$1 = \frac{A}{\mu_1-1}\frac{\hat{q}_1}{\hat{q}_2}\left(\frac{\hat{q}_2}{\hat{q}_1}\hat{u}_1 - \mu_1\hat{u}_2\right). \tag{391}$$

This condition can be solved for $\hat{u}_2$ in terms of $\hat{u}_1$,

$$\hat{u}_2 = \frac{1}{\mu_1}\frac{\hat{q}_2}{\hat{q}_1}\left(\hat{u}_1 - \frac{\mu_1 - 1}{A}\right), \tag{392}$$

which gives the self-consistency condition (from $\hat{u}_2 = \hat{u}_2$)

$$(A\hat{u}_1 + 1) - \mu_1 = \mu_1 (A\hat{u}_1 + 1) \sum_{j=2}^{L} \frac{\mu_j - 1}{\mu_j}. \tag{393}$$

This can be re-written as

$$\frac{A\hat{u}_1}{A\hat{u}_1 + 1} = \sum_{j=1}^{L} \frac{\mu_j - 1}{\mu_j}, \tag{394}$$

which gives

$$\hat{u}_1 = \frac{1}{A} \frac{\sum_{j=1}^{L} (\mu_j - 1)/\mu_j}{1 - \sum_{j=1}^{L} (\mu_j - 1)/\mu_j}. \tag{395}$$

For $\ell = 2, \ldots, L$, we can then write

$$\hat{u}_\ell = \frac{\hat{q}_\ell}{\hat{q}_1} \frac{A\hat{u}_1 + 1}{A} \sum_{j=\ell}^{L} \frac{\mu_j - 1}{\mu_j}, \tag{396}$$

hence

$$\hat{u}_\ell = \frac{\hat{q}_\ell}{\hat{q}_1} \frac{1}{A} \frac{\sum_{j=\ell}^{L} (\mu_j - 1)/\mu_j}{1 - \sum_{k=1}^{L} (\mu_k - 1)/\mu_k} \tag{397}$$

for all $\ell = 1, \ldots, L$.

In terms of the original variables, we then have

$$\hat{c}_\ell = \alpha_1 \hat{q}_1^2 c_1 \hat{u}_\ell \tag{398}$$

$$= c_1 \frac{\alpha_1 \hat{q}_1 \hat{q}_\ell}{A} \frac{\sum_{j=\ell}^{L} (\mu_j - 1)/\mu_j}{1 - \sum_{k=1}^{L} (\mu_k - 1)/\mu_k} \tag{399}$$

$$= c_1 \frac{\hat{q}_\ell}{q_1} \frac{\sum_{j=\ell}^{L} (\mu_j - 1)/\mu_j}{1 - \sum_{k=1}^{L} (\mu_k - 1)/\mu_k}. \tag{400}$$

Finally, for $\ell = 2, \ldots, L$, we obtain

$$c_\ell = \frac{q_\ell}{q_1} c_1 \frac{1}{\mu_\ell} \frac{1 - \mu_\ell \sum_{j=\ell+1}^{L} (\mu_j - 1)/\mu_j}{1 - \sum_{k=1}^{L} (\mu_k - 1)/\mu_k}, \tag{401}$$

with

$$c_L = \frac{q_L}{q_1} c_1 \frac{1}{\mu_L} \frac{1}{1 - \sum_{k=1}^{L} (\mu_k - 1)/\mu_k} \tag{402}$$

in particular. These are the results reported in §D.2.

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
