# Peer review of "Replica method for eigenvalues of real Wishart product matrices"

_SciPost Physics Core_

## Round 1 · Referee Report · Anonymous · 2022-9-29

Strengths

1. The paper provides a compact and pedagogical derivation of the spectral density and average location of extreme eigenvalues for a class of product random matrices of the Wishart type.

2. The paper is well-written, easy to follow, and the results appear to be correct

Weaknesses

1. The final results are not new - they have been derived already with a number of other techniques

2. Although the replica method has not been used (to my knowledge) for precisely this problem, the derivation is very standard by now, and one could convincingly argue that this journal is not the right forum to disseminate pedagogical but not novel material

Report

This paper reports the calculation of the spectral density, and the average location of extreme eigenvalues, for a class of product matrices of the Wishart type. The paper is well-written, adopts a nice and fresh pedagogical style, and the results appear to be correct.

Apart from a couple of notational remarks and some comments about omitted references, my main objection here is that (i) the results are not new, and (ii) the replica methodology is so standard by now, that one could almost argue in favour of the 'exercise-like' nature of this enterprise, which unfortunately directly clashes with both essential acceptance criteria for this journal, namely:

- Address an important (set of) problem(s) in the field using appropriate methods with an above-the-norm degree of originality;

- Detail one or more new research results significantly advancing current knowledge and understanding of the field.

As a curious background fact, one of the papers where the main result has been already derived (using the cavity method) is Ref. [8] from 2014 - where on pag. 5 the authors announced that the replica calculation for the same problem would be used in a forthcoming Part II ... which never materialised. I appreciate that the authors of the paper under review here should not be penalised for the 8-years long failure by the authors of Ref. [8] to keep their promise and deliver the replica calculation, but this promise (however broken) provides further evidence that already back in 2014 the replica calculation presented here was considered eminently doable (and was also probably done - just not written down).

It is of course a matter of judgment for the handling editor whether a correct (alternative) calculation of an already well-known fact using a very standard technique raises to the level of novelty/originality required for a research publication in this journal. As an alternative, one could make a case for diverting the paper towards SciPost Physics Lecture Notes - or other similar journals devoted to pedagogical derivations of known results.

Requested changes

1. Very minor point, but I found the choice of the letter R to denote the Stieltjes transform quite unusual (and not recommended). Usually, R denotes a close relative to the functional inverse of G(z) - the commonly denoted Stieltjes transform, and I would suggest that the authors stick to this very standard notation to avoid confusion

2. For the minimum and maximum eigenvalue -- which are random variables -- what is being computed using e.g. (22) is the average (or typical value) of the random variable $\lambda_{\mathrm{min/max}}$. It would be good to make this explicitly, for example wrapping it around angled brackets $\langle\cdot\rangle$.

3. Referencing [24] for the trick of solving for the minimum/maximum eigenvalue using a low-temperature expansion of an auxiliary canonical partition function is fine, but there are more to-the-point and recent references that ought to be mentioned, such as

Kabashima et al 2010 Cavity approach to the first eigenvalue problem
in a family of symmetric random sparse matrices J. Phys.: Conf. Ser. 233 012001

Kabashima et al 2012 First eigenvalue/eigenvector in sparse random symmetric matrices: influences of degree fluctuation J. Phys. A: Math. Theor. 45 325001

Vito A R Susca et al 2019 J. Phys. A: Math. Theor. 52 485002

and other references therein

  • validity: top
  • significance: ok
  • originality: low
  • clarity: top
  • formatting: perfect
  • grammar: perfect

Author:  Jacob Zavatone-Veth  on 2022-12-08  [id 3115]

(in reply to Report 1 on 2022-09-29)

Thank you for your careful reading of our manuscript. We have uploaded a PDF with a point-by-point response to your concerns; please also see our resubmitted manuscript.

Attachment:

replica_approach_to_random_gaussian_product_matrices_review__M7lUkP0.pdf

---

## Round 1 · Referee Report · Anonymous · 2022-10-12

Report

The authors show that a replica-symmetric approach can be employed to analyze the limiting eigenvalue spectrum of a real Wishart product matrix (in an appropriate thermodynamic limit). They thus recover known relations involving its resolvent and its minimum/maximum eigenvalues by means of a physicist-friendly derivation. Although these results are expressly not new, their derivation is interesting and it complements the original (more rigorous) approaches presented in Refs. [1,5].

However, before recommending the publication of this manuscript I would like the authors to address the following points, most of which concern the exposition of the method and the overall structure of the paper.

Requested changes

1) On p. 2, the authors write: "our objective in reporting this replica-theoretic derivation is to note its simplicity, as the replica method has to the best of our knowledge not seen broad application to the study of product random matrices".
On the other hand, the replica method has indeed seen broad applications in the analysis of other random matrix ensembles, and this should at least be mentioned. For instance, the seminal work of Bray-Rodgers in Phys. Rev. B 37, 3557 (1988) (on the spectrum of sparse random matrices) should be cited somewhere. More recent works by (just to name a few)
A. Cavagna & J. P. Garrahan & I. Giardina (2000),
F. Metz & G. Parisi & L. Leuzzi (2014),
F. Metz & I. Pérez Castillo (2016-2017),
explicity carry out replica calculations both for the GOE ensemble and for sparse random matrices, and it would be relevant to include them and a few others in the literature.

2) Moreover, the works recalled above generally address at least some new results (e.g., quantifying the finite size corrections) which are hardly achievable by means of other methods (e.g., the resolvent method or the orthogonal polynomial method). I'm wondering whether a slight extra effort within the replica formalism may teach us something new about the Wishart matrix product ensemble too.

Similarly, in the concluding Section 4 the authors may want to comment on why are the open problems they list supposed to be more easily addressed within the replica formalism, rather than using other standard methods.

3) By reading this manuscript, I often had the feeling that I was reading some notes rather than a paper. I believe this can be greatly improved by doing the following:
- A "plan of the paper" with the aim of clarifying its structure (usually positioned at the end of the Introduction) is currently missing. As a result, when I first read Sections 1.1 and 1.2, I was unable to tell apart the results derived in this manuscript from the "context" and known background facts that the authors were probably trying to provide.
- Sections 2 and 3 could be shortened significantly by reporting part of the calculation in one or a few appendices. I do understand that the focus of this work is on the method, and so of course the exposition of such method naturally occupies a large part of the manuscript. Nonetheless, I believe that not all the algebraic steps are strictly necessary to the comprehension of the method. The overall effect is that the important details get inevitably lost in the middle of uninteresting algebra, and the method ends up looking less "compact" than initially claimed (and than it actually is).
- Some more attention should be paid to the punctuation at the end of the equations (see e.g. Eqs.11,12,16,105,136,185). I also found a couple of (potentially confusing) typos just above Eqs. 115,118.

4) Eqs. 51-52. I could not find the definition of the vector $\mathbb{1}_m$ anywhere in the manuscript (although I could still understand what follows by supposing its definition was the same as in Ref. [14], to which the authors often refer).

5) This is very pedantic, and I apologize in advance, but the fact that the Stieltjes transform was named $R(z)$ in the manuscript (I guess for "resolvent") confused me a bit. In the random matrix theory community the resolvent is often named $G(z)$ to avoid confusion with, well, the $R$-transform in free probability theory (they are simply related by $1/G(z) + R(G(z))=z$). Changing the notation to $G(z)$ is virtually costless, while I'm sure many readers would be grateful.

  • validity: high
  • significance: good
  • originality: good
  • clarity: high
  • formatting: acceptable
  • grammar: perfect

Author:  Jacob Zavatone-Veth  on 2022-12-08  [id 3114]

(in reply to Report 2 on 2022-10-12)
Category:
answer to question

Thank you for your careful reading of our manuscript. We have attached a PDF containing a point-by-point response; please also see our resubmitted manuscript.

Attachment:

replica_approach_to_random_gaussian_product_matrices_review_response.pdf

Anonymous on 2023-01-09  [id 3218]

(in reply to Jacob Zavatone-Veth on 2022-12-08 [id 3114])

I thank the authors for their efforts, as I feel that they have satisfactorily addressed all of the issues I had raised in my report.
(I apologize for the delay , but I hadn't been notified of your resubmission due to a technical problem.)

In particular, the analysis presented in Section 2.3 of the revised manuscript is interesting, and it contains new results (to the best of my knowledge).
At this stage, the Editor may judge which one is the best venue for this work among SciPost Physics Core and SciPost Physics Lecture Notes -- I am happy with both.

---

## Editorial Decision

resubmitted